# From RNNs to Transformers: Benchmarking deep learning architectures for hydrologic prediction

Jiangtao Liu[1*], Chaopeng Shen[1], Fearghal O'Donncha[2], Yalan Song[1], Wei Zhi[3], Hylke E. Beck[4], Tadd Bindas[1], Nicholas Kraabel[1], Kathryn Lawson[1]

[1] Civil and Environmental Engineering, The Pennsylvania State University, University Park, PA, USA

[2] IBM Research, Dublin, Ireland

[3] Hohai University, Nanjing, China

[4] King Abdullah University of Science and Technology, Thuwal, Saudi Arabia

* *Correspondence to*: Jiangtao Liu (jql6620@psu.edu)

**Abstract.** Recurrent Neural Networks (RNNs) such as Long Short-Term Memory (LSTM) have achieved significant success in hydrological modeling. However, the recent breakthroughs of foundation models like ChatGPT and the Segment Anything Model (SAM) in natural language processing and computer vision have raised interest in the potential of attention mechanism-based models for hydrologic predictions. In this study, we propose a deep learning framework that seamlessly integrates multi-source, multi-scale data and multi-model modules, creating an automated platform for multi-dataset benchmarking and attention-based model comparisons beyond LSTM-centered tasks. The proposed framework enables evaluation of deep learning models across diverse hydrologic prediction tasks, including regression (daily runoff, soil moisture, snow water equivalent, and dissolved oxygen prediction), forecasting (using lagged hydrologic observations combined with meteorological inputs), autoregression (forecasting based solely on historical observations), spatial cross-validation (assessing model generalization to ungauged regions), and zero-shot forecasting (prediction without task-specific training data). Specifically, we benchmarked 11 Transformer-based architectures against a baseline Long Short-Term Memory (LSTM) model and further evaluated pretrained Large Language Models (LLMs) and Time Series Attention Models (TSAMs) regarding their capabilities for zero-shot hydrologic forecasting. Results show that LSTM models perform best in regression tasks, especially on the global streamflow dataset (median KGE = 0.75), surpassing the best-performing Transformer-based model's KGE value by 0.11. However, as tasks become more complex (from regression and forecasting to autoregression and zero-shot prediction), attention-based models gradually surpass LSTM models. This study provides a robust framework for comparing and developing different model structures in the era of large-scale models, providing a valuable benchmark for water resource modeling, forecasting, and management.

## 1. Introduction

Accurate prediction of hydrologic variables such as streamflow and soil moisture is critical for various applications including agricultural irrigation planning (Zhang et al., 2021), flood management (Basso et al., 2023), landslide susceptibility assessment (Liu et al., 2025), and ecosystem conservation (Aboelyazeed et al., 2023, 2025). With the increasing frequency of extreme events, there is a growing demand for reliable and precise prediction approaches (Basso et al., 2023; Zhang et al., 2023). However, developing a unified deep learning framework that can capture and simulate multiple hydrologic variables remains challenging because hydrologic systems are inherently complex, uncertain, and often suffer from limited data availability (Reichstein et al., 2019; Shen, 2018).

In recent years, deep learning (DL) models — particularly Long Short-Term Memory (LSTM) networks (Hochreiter and Schmidhuber, 1997) — have achieved significant advances in hydrological modeling. They have been successfully applied to various tasks such as streamflow prediction (Feng et al., 2020, 2022; Kratzert et al., 2018), soil moisture simulation (Fang et al., 2017; Liu et al., 2022a, 2023), snow water equivalent estimation (Song et al., 2024), and water quality analysis (Zhi et al., 2021, 2023, 2024). Meanwhile, the advancements in Natural Language Processing and Computer Vision technologies, such as ChatGPT (OpenAI, 2023), have demonstrated the transformative potential of attention-based methods. Inspired by these advances, researchers have started exploring Transformer architectures (Vaswani et al., 2017) in hydrological modeling (Castangia et al., 2023; Koya and Roy, 2024; Liu et al., 2024a; Pölz et al., 2024; Yin et al., 2023). These studies aim to explore the potential of Transformer models to capture complex dependencies and enhance performance in hydrologic predictions.

Due to Transformers' flexible attention-based mechanisms, they can capture dependencies among sequence features. Transformers are well suited for handling multivariate data (Wu et al., 2022) and multi-scale temporal features (Gao et al., 2023). Additionally, their modular framework provides new opportunities for interpretability (Orozco López et al., 2024) and integrating with physical hydrological models (Geneva and Zabaras, 2022). Despite this potential, there are still many challenges in applying them to hydrological applications. Training Transformers usually requires significant computational resources and large datasets (Liu et al., 2024a). However, hydrologic data typically exhibit nonstationary and heterogeneous characteristics (Kirchner, 2024), limiting the availability of datasets. In order to fully utilize Transformers' capabilities, it is necessary to investigate the performance of different architectures across different hydrological datasets.

Currently, the relative performance between LSTM and Transformer models in hydrologic prediction remains an ongoing debate. For example, Pölz et al. (2024) reported that Transformer models performed very well in karst spring prediction over a four-day forecast horizon, achieving an accuracy improvement of 9% compared to LSTM models. They also observed better performance from Transformers during the snowmelt period. In contrast, Liu et al. (2024a) showed that the standard Transformer performed worse than LSTM, with only a modified Transformer achieving comparable performance. Such discrepancies have been observed in other fields, including financial time series forecasting, where attention-based models sometimes underperform LSTM-based models (Bilokon and Qiu, 2023). These inconsistent results indicate that model performance may depend on specific features, such as data scales (Ghobadi and Kang, 2022), the variables being predicted (Sun et al., 2024), and the type of task. Consequently, developing a unified modeling framework, along with consistent datasets and standardized evaluation criteria, is important for systematically assessing various DL models across different hydrologic tasks.

In addition to model evaluation challenges, prediction in ungauged basins (PUB) remains an important concern within the hydrologic community (Feng et al., 2020, 2023; Ghaneei et al., 2024). In regions such as Africa and the Tibetan Plateau, observational data is extremely limited due to remote locations, insufficient infrastructure, and high costs of gauging station installation. These factors constrain the application of DL methods (Bai et al.,

2016; Jung et al., 2019; Liu et al., 2018). To address such data limitations, zero-shot learning approaches have emerged as potential solutions. Zero-shot learning refers to the capability of a model to make predictions for unseen tasks or new types of data without additional training or fine-tuning using target-specific data (Gruver et al., 2024; Wang et al., 2019). For example, pre-trained Large Language Models (LLMs) (Tan et al., 2024; Zhang et al., 2024) and Time Series Attention-based Models (TSAMs) (Ansari et al., 2024; Ekambaram et al., 2024; Rasul et al., 2024) can transfer generalized knowledge learned from non-hydrologic contexts to hydrologic prediction in ungauged basins. This differs from transfer learning, which requires re-training or fine-tuning using data from the target basin (Ma et al., 2021; Pham et al., 2023). Leveraging the generalization capabilities of LLMs and TSAMs, zero-shot methods may offer predictive performance comparable to supervised methods, representing a promising solution for hydrologic forecasting in data-scarce regions.

Current deep learning frameworks in hydrology often focus on streamflow prediction using LSTM models and lack extensive support for multi-dataset benchmarking as well as comparisons among attention-based architectures. To bridge this gap, we propose a systematic multi-source hydrologic modeling framework. This framework provides plug-and-play integration of various models, flexible task configuration, and end-to-end automation from data processing to model training and validation. Additionally, it combines LLMs and TSAMs to improve model applicability in regions with limited data availability. Note that this study focuses on data-driven models. Integration with physics-informed differentiable models will be explored in future research (Feng et al., 2022; Song et al., 2025; Tsai et al., 2021). Through this study, we aim to address the following three scientific questions:

1. Can a single deep learning framework tackle myriad tasks ranging from soil moisture and streamflow to water chemistry and snow water equivalent, to enable comparisons among different model architectures?
2. How do various attention-based architectures perform compared to LSTM models across tasks with varying complexity?
3. To what extent can large, pre-trained models (e.g., LLMs or TSAMs) be applied in ungauged basins?

## 2. Data and models

### 2.1. Overview

Our framework comprises three core components: data processing, model management, and task execution (Supplementary Fig. S1). In the data processing module, raw data from various formats (e.g., TXT, CSV, GeoTIFF) are first converted into NetCDF files for consistent handling. The model management module provides a unified interface to manage different deep learning models, including Transformers and benchmark models. The task execution module enables users to select and run different hydrologic tasks (e.g., regression, forecasting).

### 2.2. Datasets

We mainly used five multi-source hydrologic datasets (Supplementary Fig. S2), including two runoff datasets—Catchment Attributes and Meteorology for Large-sample Studies (CAMELS) and Global Streamflow—as well as datasets for soil moisture (ISMN), snow water equivalent (SWE), and dissolved oxygen (DO) (Table 1).

The CAMELS dataset (Addor et al., 2017; Newman et al., 2014) served as our benchmark dataset for runoff modeling across the conterminous United States (CONUS). CAMELS includes static attributes and dynamic

forcing data from Daymet (Thornton et al., 1997), Maurer (Maurer et al., 2002), and the North American Land Data Assimilation System (NLDAS) (Xia et al., 2012). To ensure a fair comparison with previous studies, we used the same 531 basins and data processing methods as in Kratzert et al. (2021).

We used a global dataset compiled by Beck et al. (2020), who initially collected daily observed streamflow data from 21,955 basins listed in multiple national and international databases. After excluding basins with incomplete daily records and non-reference basins, Beck et al. (2020) retained catchments between 50 km$^2$ (to ensure sufficient spatial resolution of gridded meteorological inputs) and 5,000 km$^2$ (to minimize channel routing and reservoir operation effects at daily scales), resulting in a total of 4,299 basins. Based on that subset, we conducted additional manual quality checks by visually inspecting streamflow time series for each basin. Basins exhibiting flat lines, abrupt discontinuities, or very short records (spanning only a few months) were excluded, ultimately resulting in 3,434 basins usable for our analyses.

The global soil moisture dataset from the International Soil Moisture Network (ISMN) (Dorigo et al., 2011, 2013; Liu et al., 2023) includes observations from 1317 sites. It provides forcing variables, terrain attributes, soil attributes, and land cover information for analyses.

The snow water equivalent (SWE) data used in our study follows Song et al. (2024), who utilized SNOTEL observations (USDA NRCS NWCC, 2021) from 525 sites across regions in the western United States. Meteorological forcing data include precipitation, air temperature, downward shortwave radiation, wind speed, and humidity. In addition, static attribute data (e.g., station latitude, elevation, aspect, and land cover) were used to provide contextual information.

We used dissolved oxygen data following Zhi et al. (2021), based on the CAMELS-Chem (Sterle et al., 2020) dataset. CAMELS-Chem compiles USGS water chemistry and streamflow records from 1980 to 2014 across the United States. Although the original CAMELS-Chem includes 506 basins, Zhi et al. selected a subset of 236 basins, each having at least 10 dissolved oxygen measurements.

All datasets used in this study have a daily temporal resolution. For the CAMELS and Global Streamflow datasets, the target variables were daily observed streamflow at basin outlets. Dynamic meteorological forcings (e.g., precipitation, temperature) and static attributes (e.g., elevation, soil properties) were obtained by spatially averaging gridded data over each catchment. For the ISMN (soil moisture) and SWE datasets, target variables were point-based observations, and their corresponding dynamic and static predictors were directly extracted from gridded datasets at the geographic coordinates of each observation site. The DO dataset, similar to CAMELS, contains target variables measured at basin outlets, with basin-averaged dynamic and static inputs derived from gridded data. Latitude and longitude coordinates were used solely for spatial extraction of predictors and were not included as direct input features for model training, except for latitude in the SWE dataset.

**Table 1. Overview of datasets.**

| Dataset | Period (Training / Validation / Testing) | Target Variable | Dynamic Variables | Static Variables |
|---|---|---|---|---|
| CAMELS | 1999-10-01 to 2008-09-30 / 1980-10-01 to 1989-09-30 / 1989-10-01 to 1999-09-30 | Streamflow | precipitation, solar radiation, maximum temperature, minimum temperature, vapor pressure (NLDAS, Maurer, Daymet) | elevation, slope, area, forest fraction, leaf area index (LAI), green vegetation fraction (GVF), soil depth, porosity, conductivity, water content, soil texture fractions (sand, silt, clay), carbonate rock fraction, permeability, climate indices (mean precipitation, potential evapotranspiration (PET), aridity, snow fraction, precipitation frequency and duration extremes) |
| Global Streamflow | 1999-01-01 to 2016-12-31 / 1998-01-01 to 1998-12-31 / 1980-01-01 to 1997-12-31 | Streamflow | precipitation, PET, maximum temperature, minimum temperature | mean precipitation, seasonality precipitation, seasonality PET, snow fraction, snowfall fraction, mean temperature, normalized difference vegetation index (NDVI), elevation, slope, aspect, soil texture fractions (sand, silt, clay), soil depth, permeability, porosity, carbonate rock fraction, forest fraction, grassland fraction, soil erosion, area |
| ISMN | 2017-01-01 to 2020-12-31 / - / 2015-04-01 to 2016-12-31 | Soil Moisture | soil temperature, surface pressure, solar radiation, air temperature, evaporation, wind speed, volumetric soil water, precipitation | elevation, slope, aspect, soil texture (sand, clay, silt, bulk density), land surface temperature, albedo, landcover, NDVI, profile curvature, roughness, mean Soil Moisture Active Passive Data (SMAP) |
| SWE | 2001-01-01 to 2015-12-31 / - / 2016-01-01 to 2019-12-31 | Snow Water Equivalent | precipitation, maximum temperature, minimum temperature, solar radiation, wind speed, humidity | latitude, elevation, slope, aspect, land cover, forest fraction, root depth, soil depth, porosity, permeability |
| DO | 1980-01-01 to 2000-12-31 / - / 2001-01-01 to 2014-12-30 | Dissolved Oxygen | precipitation, solar radiation, maximum temperature, minimum temperature, vapor pressure, streamflow | elevation, slope, area, forest fraction, LAI, GVF, soil depth, porosity, conductivity, water content, soil texture fractions (sand, silt, clay), carbonate rock fraction, permeability, climate indices (mean precipitation, PET, aridity, snow fraction, precipitation frequency and duration extremes) |

**2.3. Attention models**

Our framework integrates a total of 13 models, including 11 attention-based architectures (Table 2): CARDformer (originally CARD) (Xue et al., 2024), Crossformer (Zhang and Yan, 2022), ETSformer (Woo et al., 2022), Informer (Zhou et al., 2021), iTransformer (Liu et al., 2024b), Non-stationary Transformer (Liu et al., 2022b), Pyraformer (Liu et al., 2021), Reformer (Kitaev et al., 2020), Vanilla Transformer (Vaswani et al., 2017), PatchTST (Nie et al., 2023) and TimesNet (Wu et al., 2022). We also included two baseline models, DLinear

(Zeng et al., 2022), and LSTM. Detailed descriptions of all deep learning models and their principles are available in the Supplementary Information (Text S1).

It is important to note that although the computational speed of PatchTST and TimesNet is acceptable when applied to a small number of catchments or stations, our experiments revealed that their training times increase dramatically as the number of basins increases, especially for regression tasks. A detailed comparison of computational times across all models is provided in the Supplementary Information (Table S7). Therefore, PatchTST and TimesNet models were only applied in the autoregression scenario within our study. In addition, we included pre-trained LLMs and TSAMs for zero-shot forecasting, as described in detail in Section 2.4.5.

**Table 2 Models evaluated in this study.**

| Model name | Main Feature | General Feature | Reference |
|---|---|---|---|
| CARDformer | Channel-aligned attention; token blend module (originally referred to as CARD in the paper) | Multivariate correlation modeling | (Xue et al., 2024) |
| Crossformer | Cross-dimension dependency; dimension-segment-wise embedding; two-stage attention | Multivariate dependency modeling | (Zhang and Yan, 2022) |
| ETSformer | Exponential Smoothing Attention (ESA); Frequency Attention (FA) | Trend and seasonality modeling | (Woo et al., 2022) |
| Informer | ProbSparse self-attention; self-attention distilling; generative style decoder | Efficient long-sequence forecasting | (Zhou et al., 2021) |
| iTransformer | Inverted Dimension; embedding the whole series as the token | Multivariate interaction modeling | (Liu et al., 2024b) |
| Non-stationary Transformer | Series Stationarization; de-stationary attention | Handling non-stationary time series | (Liu et al., 2022b) |
| PatchTST | Patch-based tokenization; channel independence | Segmentation-based time series modeling | (Nie et al., 2023) |
| Pyraformer | Pyramidal attention; hierarchical multi-resolution structure | Multi-scale temporal analysis | (Liu et al., 2021) |
| Reformer | Locality-Sensitive Hashing (LSH) attention; reversible residual layers | Efficient handling of long sequences | (Kitaev et al., 2020) |
| TimesNet | Converts 1D variations to 2D; intraperiod/interperiod analysis | Multi-periodicity pattern extraction | (Wu et al., 2022) |
| Vanilla Transformer | Multi-head self-attention; residual connections and layer normalization; positional encodings | General-purpose sequence modeling | (Vaswani et al., 2017) |
| DLinear | Trend–seasonality decomposition with linear layers | Linear modeling with explicit trend handling | (Zeng et al., 2022) |
| LSTM | Gating mechanisms (input, forget, and output gates) | Captures temporal dependencies | (Hochreiter and Schmidhuber, 1997) |

*The Vanilla Transformer used here differs from the one in Liu et al. (2024a), which applied random masking to attention weights.*

### 2.4. Task scenarios

We designed several hydrologic tasks, ranging from simple regression and forecasting (time-lagged prediction) to more complex scenarios such as autoregression and zero-shot (forecasting without training on target data). The following sections detail these tasks and their experimental setups.

### 2.4.1. Regression

The regression experiment aimed to predict a target variable within a given period, using input variables and static attributes from the same time window, without including the target variable itself as input. This task is similar to the "fill-in-the-blank" strategy in natural language processing, where models infer missing tokens based on known context. It is formally defined as:

$$y_t = f(X_{1:t}, S) \tag{1}$$

where $y_t$ represents the target variable at time $t$, $X_{1:t}$ is the input time series with a fixed length of 365 days (selected to capture annual cycles) (Fang et al., 2017; Feng et al., 2020; Kratzert et al., 2019), and $S$ indicates static attributes (e.g. basin area, elevation). We fixed the output horizon at 1 day.

### 2.4.2. Forecasting

Hydrologic variables often show strong temporal dependencies (Delforge et al., 2022). The forecasting method, also referred to as time-lagged prediction or data integration, is frequently used in hydrologic forecasting (Fang and Shen, 2020; Feng et al., 2020). In this study, forecasts were evaluated for lead times of 1, 7, 30, and 60 days. This approach combines meteorological inputs with lagged target variables to predict future target values:

$$y_{t+1:t+\rho} = f(X_{t+1:t+\rho}, y_{t-\rho+1:t}, S) \tag{2}$$

where $\rho$ represents the prediction horizon (lead time), $X_{t+1:t+\rho}$ represents the meteorological inputs for the forecast period, and $y_{t-\rho+1:t}$ contains the historical observations of the target variable. Forecast accuracy largely depends on the reliability of the meteorological forecasts. As the forecast horizon (lead time) increases, the uncertainty in the meteorological inputs usually increases as well (Wessel et al., 2024).

### 2.4.3. Autoregression

Autoregression tasks utilize historical time series data to predict future hydrologic variables. Unlike the forecasting scenario, we do not incorporate future meteorological forecasts here. Instead, models extrapolate historical trends for short- to long-term predictions. We experimented with forecast horizons of 1, 7, 30, and 60 days, representing short- to long-range forecasts:

$$y_{t+1:t+\rho} = f(y_{1:t}) \text{ or } f(y_{1:t}, S) \tag{3}$$

### 2.4.4. Spatial cross-validation

Spatial cross-validation evaluates the model's ability to generalize from training locations to new, unknown regions, thus measuring its performance at locations not included in the training set. (Liu et al., 2023). To avoid redundancy and excessive computational costs, we conducted this experiment only on the CAMELS dataset. We

randomly divided it into three spatially non-overlapping folds, each serving once as the test set, while the remaining two folds were used for training. This process was repeated for a total of three rounds, and performance metrics were averaged across these folds:

$$P_s = \frac{1}{3} \sum_{k=1}^{3} f\left(M_{train}(D_i), D_j\right)$$

(4)

where $P_s$ represents the spatial cross-validation performance metric, $M_{train}(D_i)$ is the model trained on subset $D_i$, and $D_j$ is the testing subset for fold k ($i \neq j$).

**2.4.5. Zero-Shot**

Zero-shot forecasting refers to making predictions using models that have not been trained on hydrologic data. For example, LLMs can be prompted via structured queries containing historical time series and domain context, enabling them to generate forecasts without fine-tuning. A complete prompt example is provided in Supplementary Text S2. We evaluated several LLMs via API, including GPT-3.5, GPT-4-turbo, Gemini 1 pro, Llama3 8B, and Llama3 70B (Google, 2023; Grattafiori et al., 2024; OpenAI, 2023). To ensure consistency and comparability between these models, we maintained the same parameters (e.g., temperature) and used the same prompt for each model. Specifically, historical streamflow observations (e.g., previous 90 days) and basin characteristics, such as geographic coordinates and catchment area, were converted into structured textual prompts provided to the LLMs. The textual outputs from these models were then automatically parsed into numerical forecasts using regular expression extraction (Supplementary Text S2). Although fine-tuning LLMs specifically for hydrologic modeling was not explored in this study, prompt engineering, parameter-efficient tuning methods, and systematic exploration of repeated queries and parameter variations represent promising directions for future research.

This zero-shot method can be extended to pre-trained Time Series Attention Models, such as TimeGPT (Garza et al., 2024), Lag-Llama (Rasul et al., 2024), and Tiny Time Mixers (TTMs) (Ekambaram et al., 2024). Although these models have not been trained on hydrologic data, they can still provide forecasts if supplied with historical information:

$$y_{t+1:t+rho} = f_0(y_{1:t})$$

(5)

where $f_0$ represents the pretrained model (LLM or TSAM) that has not been trained or fine-tuned for hydrologic datasets.

**2.5. Evaluation metrics**

We evaluated model performance using four metrics. For Kling-Gupta Efficiency (KGE) (Gupta et al., 2009), values approaching 1 are preferred:

$$KGE = 1 - \sqrt{(\gamma - 1)^2 + (\alpha - 1)^2 + (\beta - 1)^2}$$

(6)

$\gamma$ is the Pearson's correlation coefficient between simulated $\hat{y}$ and observed $y$, $\alpha$ is the ratio of the standard deviations of simulated to observed values, and $\beta$ is the ratio of the means of simulated to observed values.

For Nash-Sutcliffe Efficiency (NSE) (Nash and Sutcliffe, 1970), values approaching 1 are preferred:

$$NSE = 1 - \frac{\sum_{t=1}^{T}(y_t - \hat{y}_t)^2}{\sum_{t=1}^{T}(y_t - \bar{y}_t)^2} \tag{7}$$

$y_t$ and $\hat{y}_t$ are the observed and simulated values at time $t$, respectively. $\bar{y}$ is the mean of observed data.

Coefficient of determination ($R^2$) is defined here as the squared Pearson's correlation coefficient, and values approaching 1 are preferred:

$$R^2 = \left( \frac{\sum_{t=1}^{T}(y_t - \bar{y})(\hat{y}_t - \bar{\hat{y}})}{\sqrt{\sum_{t=1}^{T}(y_t - \bar{y})^2}\sqrt{\sum_{t=1}^{T}(\hat{y}_t - \bar{\hat{y}})^2}} \right)^2 \tag{8}$$

For unbiased root-mean-square error (ubRMSE), values approaching 0 are preferred:

$$ubRMSE = \sqrt{RMSE^2 - Bias^2} \tag{9}$$

$RMSE$ is the root-mean-square error between simulated and observed values. $Bias$ is the mean of simulated minus observed values.

In addition to these four metrics, we report two flow-specific metrics to assess model performance at extreme flow conditions. FHV quantifies the percent deviation of the highest 2% of flows, and FLV quantifies the percent deviation for the lowest 30% of flows (Feng et al., 2020; Yilmaz et al., 2008). Although initially developed for flow analysis, metrics conceptually similar to FHV/FLV have also been applied to other variables to evaluate model performance at their extreme upper and lower ranges (Bayissa et al., 2021; Brunner and Voigt, 2024).

### 2.6 Experimental setup and hyperparameter configuration

In the temporal experiments, each dataset was temporally divided into training, validation, and testing subsets (Table 1) to avoid information leakage. Additionally, to assess the model's spatial generalization capability, we conducted spatial cross-validation experiments. Our baseline LSTM configuration closely follows that of Kratzert et al. (2021). Prior to our main experiments, we confirmed that this baseline LSTM achieved a performance (median KGE $\approx$ 0.80) comparable to that reported by Kratzert et al. (2021), thus providing indirect validation and enabling comparability with established benchmarks.

Given the extensive experimental scale involving multiple models and datasets, we adopted previously used hyperparameters for the LSTM model from prior hydrological modeling studies (Feng et al., 2020, 2024; Kratzert et al., 2021; Liu et al., 2022a, 2023, 2024a; Song et al., 2024; Zhi et al., 2021). For Transformer-based models, hyperparameter tuning (e.g., number of encoder layers, attention heads) was conducted using the validation subset of the CAMELS dataset, guided by recommendations from the Time Series Library (Wang et al., 2024; Wu et al., 2022).

## 3. Results and discussion

### 3.1 Regression tasks

In the regression tasks, the LSTM model performed better overall compared to other models across most datasets (Fig. 1, Supplementary Table S1). For example, the LSTM model achieved KGE values of 0.80 for CAMELS, 0.75 for global streamflow, 0.71 for soil moisture, and 0.70 for dissolved oxygen. However, for the snow water equivalent dataset, the Non-stationary Transformer model slightly outperformed LSTM, achieving a higher KGE value of 0.88 compared to 0.87 for LSTM. This result suggests that the LSTM model may be nearing its performance ceiling, consistent with previous studies (Liu et al., 2024a; Vu et al., 2023) reporting minimal performance differences between Transformer variants and LSTM. These observations imply that additional improvements may be inherently limited by data uncertainties or intrinsic constraints of current hydrologic datasets. On the CAMELS and global streamflow datasets, the LSTM model achieved the highest KGE values (0.80 and 0.75, respectively). Crossformer ranked second on the CAMELS dataset, falling behind LSTM by 0.07, while Informer was second-best on the global streamflow dataset, trailing LSTM by 0.11. Performance gaps across datasets varied, ranging from a narrow margin of 0.01 on dissolved oxygen to a more substantial difference of 0.11 on global streamflow.

As the dataset scale increased, the advantages of LSTM in reducing overall prediction error became more pronounced. For instance, on the US-scale (CAMELS) and global streamflow datasets, LSTM reduced median ubRMSE by 11.3% and 12.0%, respectively, compared to the best-performing attention-based models. Despite LSTM performing well in minimizing overall errors, Transformers sometimes performed better at capturing extreme values, including both high and low flow conditions. For example, the Non-stationary Transformer model performed better under high-flow conditions, achieving an FHV score of -4.10 for global streamflow predictions. Moreover, attention-based models outperformed the LSTM model in capturing low extremes (FLV metric) for both dissolved oxygen and global soil moisture datasets, achieving FLV values of 0.15 and 0.99, respectively. These results suggest that attention-based models may offer advantages in specialized tasks involving complex and extreme events.

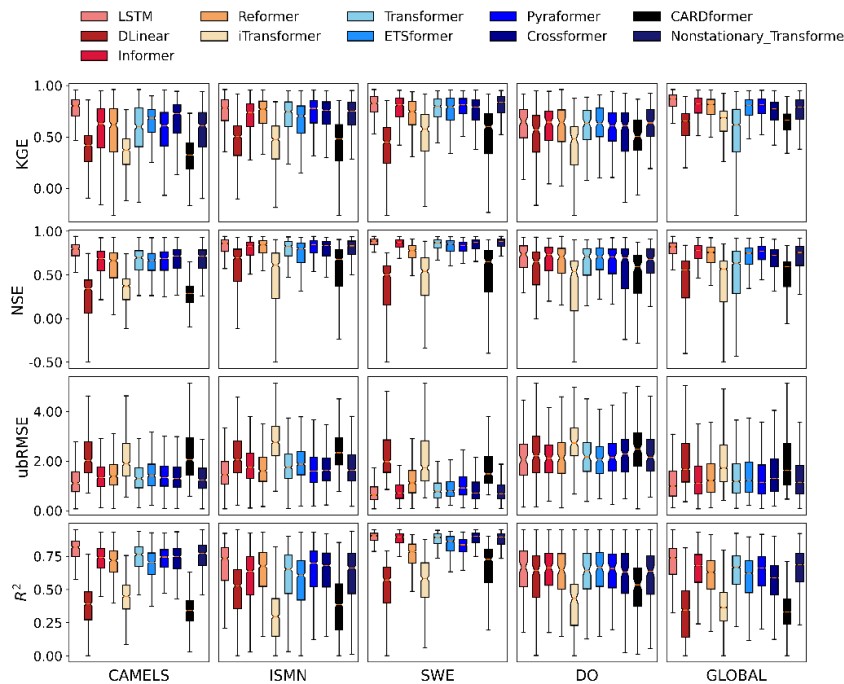

**Figure 1. Performance comparison of 11 models across five datasets for regression tasks. The horizontal axis represents the five datasets: CAMELS (conterminous United States (CONUS) scale streamflow), ISMN (global-scale soil moisture), SWE (snow water equivalent in the western United States), DO (dissolved oxygen at the CONUS scale), and GLOBAL (global-scale streamflow). The vertical axis displays four evaluation metrics: Kling–Gupta Efficiency (KGE; values approaching 1 are desired), Nash–Sutcliffe Efficiency (NSE; values approaching 1 are desired), unbiased Root-Mean-Square Error (ubRMSE; values approaching 0 are desired), and Coefficient of Determination ($R^2$; values approaching 1 are desired). Each box plot shows the distribution of the model's performance for a specific dataset–metric combination, with horizontal lines indicating median values. Detailed numerical results are provided in Supplementary Table S1.**

Beyond overall performance metrics, it is important to examine how these models behave across different geographic locations. We observed a phenomenon called "spatial amplification effect", where smaller basins (less than 500 km²) exhibit a stronger rainfall-runoff response. This pattern aligns with the principle of "wet-gets-wetter, dry-gets-drier", meaning regions already experiencing a lot of precipitation tend to become wetter, while dry areas become drier (Faghih and Brissette, 2023; Hogikyan and Resplandy, 2024). This spatial amplification effect became apparent when comparing the spatial distributions of model performance between LSTM and attention-based models. Attention-based models tended to improve simulations in regions where performance was already good but conversely degraded results in areas with pre-existing challenges. To explore these patterns, we developed an interactive visualization website (https://attention-lstm-difference-11f95e692628.herokuapp.com/). Users can select a baseline model, a comparison model, a metric, and a dataset to visualize spatial performance maps and their differences. For example, in streamflow simulation using the CAMELS dataset, the attention-based models performed well in the eastern and coastal western United States, where the LSTM models also demonstrated strong results. Similarly, the spatial amplification effect was observed in SWE predictions made by the Non-stationary Transformer, which outperformed LSTM in mountainous regions such as California's

Eldorado National Forest and Oregon. However, the Non-stationary Transformer underperformed in the challenging regions between New Mexico and Arizona. Future work can investigate methods for selecting suitable models based on local characteristics or combine multiple models into an ensemble to leverage their respective strengths.

### 3.2. Forecasting task

In the forecasting tasks using the CAMELS dataset for streamflow prediction, we evaluated model performance at lag intervals of 1, 7, 30, and 60 days. At a short lag interval (1 day), LSTM performed the best, achieving a KGE value of 0.89, and a ubRMSE value of 0.91 (Fig. 2, Supplementary Table S2). By comparison, the best-performing attention-based model (ETSformer) had a lower performance, with a KGE of 0.81 and ubRMSE of 1.09. However, as the prediction interval increased, the performance difference between LSTM and attention-based models gradually decreased. At a 1-day lag, LSTM outperformed the best attention-based model by approximately 0.08 in terms of KGE, but this advantage narrowed to 0.01 with a 30-day lag. Some attention-based models, such as the Non-stationary Transformer model, closely matched the performance of LSTM, achieving a KGE of 0.81 with a 7-day lag, and 0.82 with a 30-day lag. The ubRMSE values of these attention-based models also remained within approximately a 10% margin of those obtained by LSTM. Overall, while LSTM outperformed attention models in short-range forecasting, attention models became more competitive with longer time lags, especially in situations where the LSTM had high uncertainty.

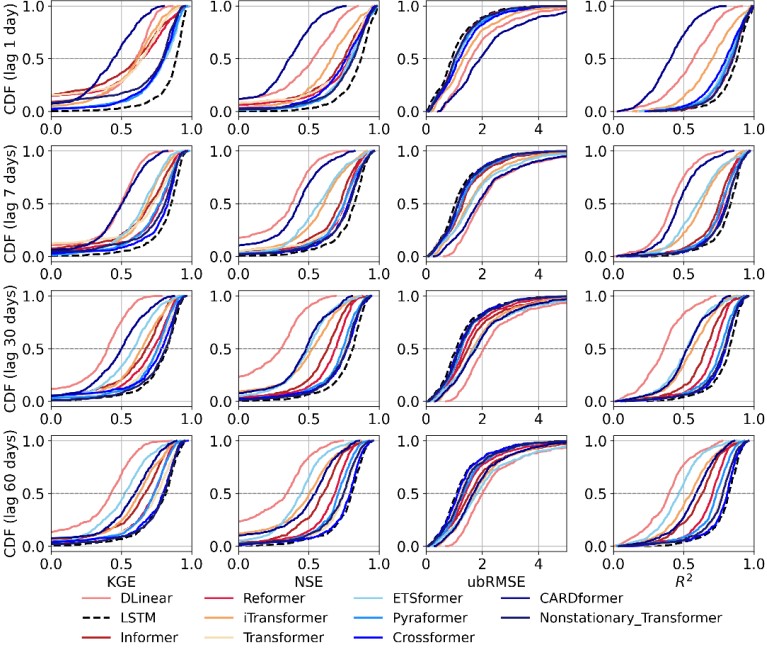

**Figure 2. Comparative analysis of the Cumulative Density Functions (CDF) from time-lagged forecasting experiments. Each CDF represents the distribution of a model's performance metrics across all basins. The four columns correspond to different evaluation metrics: Kling–Gupta Efficiency (KGE), Nash–Sutcliffe Efficiency (NSE), unbiased Root-Mean-Square Error (ubRMSE), and Coefficient of Determination ($R^2$). The four rows represent different time lags: 1 day, 7 days, 30 days, and 60 days. Horizontal lines indicate median values (CDF = 0.5). Detailed numerical results, including median values, are provided in Supplementary Table S2.**

Some attention-based models showed an advantage over LSTM in simulating either high flow or low flow conditions, though performance varied depending on the chosen model and hydrological conditions. For example, the Non-stationary Transformer displayed lower FHV values (e.g., -0.42, and -5.22 for 1-, and 30-day lags, respectively), indicating a reduced bias when predicting high-flow conditions. Other attention-based models, such as Pyraformer and Reformer, performed well at low-flow predictions. Pyraformer, for example, achieved FLV values of -3.39 at a 1-day lag and 1.6 at a 7-day lag, both surpassing LSTM. These differences highlight the capability of the attention mechanism to dynamically assign weights to different parts of the input sequence and capture critical variations relevant to flow extremes.

### 3.3. Autoregression tasks

All models performed best when forecasting for one day ahead, primarily due to the strong autocorrelation in daily runoff data (Fang et al., 2017; Feng et al., 2020). In this short-term forecasting scenario, attention-based models slightly outperformed LSTM in terms of the KGE metric (Fig. 3, Supplementary Table S3). For example, Pyraformer achieved a KGE value of 0.65, slightly higher (by 0.01) than LSTM. The 1-day forecast represents the simplest scenario, because high performance can be achieved by simply setting the forecast equal to the previous day's runoff. However, as the forecasting horizon increased to longer periods (e.g., 7, 30, and 60 days), attention-based models began to substantially outperform LSTM. LSTM's KGE dropped from 0.15 with a 7-day horizon to -0.03 with a 30-day horizon. Its $R^2$ value also decreased from 0.12 to 0.03, and finally to 0.01. In contrast, attention-based models maintained relatively stable performances at longer horizon days. For example, at the 7-day forecast horizon, Pyraformer, PatchTST, and Crossformer all achieved KGE values nearly twice that of LSTM. In particular, Pyraformer obtained a KGE value of 0.32 and an $R^2$ value of 0.23. At forecast horizons of 30 and 60 days, TimesNet, Reformer, and Pyraformer showed relatively better performance compared to LSTM, suggesting the attention mechanism's potential ability to better capture long-term temporal patterns, although absolute performance still remained limited.

To examine whether auxiliary information could improve model performance, we introduced static attribute data as additional inputs (Table S4). Consistent with our earlier findings, models continued to perform strongly at the shortest (1-day) forecasting horizon, benefiting from the auxiliary data. For example, LSTM's KGE value improved from 0.64 to 0.81 (an increase of 0.17), and its $R^2$ value improved from 0.59 to 0.79 (an increase of 0.2). Meanwhile, its ubRMSE decreased by 29.3% (Tables S3). Nevertheless, as the forecasting horizon increased (to 7, 30, and then 60 days), the performance of LSTM dropped significantly, with KGE values of only 0.15, 0.02, and -0.04, respectively. This decline likely stems from the inherent limitations of RNN-based models in processing long sequences, particularly under highly nonstationary conditions. In contrast, attention-based models such as ETSformer, Pyraformer, and Crossformer also benefited from incorporating auxiliary data, achieving KGE performance improvements ranging from 0.01 to 0.15. Nevertheless, even the best attention model achieved KGE and $R^2$ values below 0.5 at longer-term forecasting horizons, highlighting the challenges deep learning models face in providing accurate long-range predictions. Overall, incorporating auxiliary information was beneficial for forecasting but offered only limited improvements at longer forecast horizons.

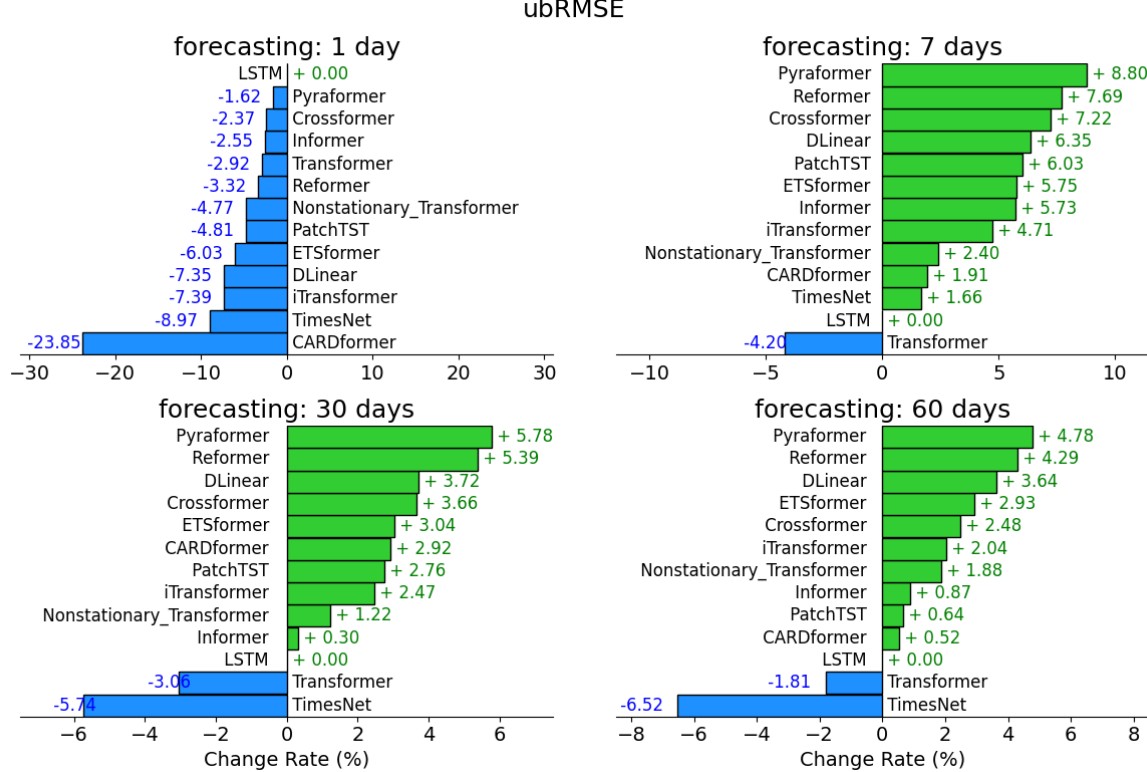

**Figure 3. Comparison of percent changes in ubRMSE for various models at four forecasting horizons: 1 day (top left), 7 days (top right), 30 days (bottom left), and 60 days (bottom right). Each subplot shows the percent changes relative to the LSTM baseline (set at 0%), calculated as: (ubRMSE_LSTM - ubRMSE_model)/ubRMSE_LSTM × 100%. Positive changes (green) indicate models with smaller ubRMSE values (better performance) compared to LSTM, while negative changes (blue) indicate models with larger ubRMSE values (worse performance). Detailed numerical results are provided in Table S3 (minor differences may exist due to rounding).**

### 3.4. Model generalization for Prediction in Ungauged Basins (PUB)

When shifting from temporal predictions to spatial cross-validation experiments, all models experienced performance declines. However, our results showed that attention-based models had relatively smaller decreases compared to LSTM (Fig. 4, Supplementary Table S5). For example, in spatial cross-validation on the CAMELS dataset, the KGE of LSTM dropped from 0.80 to 0.62, whereas that of Crossformer dropped from 0.73 to 0.63. Crossformer thus slightly outperformed LSTM under this spatial generalization scenario. Attention-based models continued to outperform the LSTM model in high-flow predictions. For example, Crossformer's FHV showed an improved FHV of -6.76 compared to LSTM (-14.13). Looking at performance another way, CARDformer and iTransformer yielded relatively modest $R^2$ values of 0.4. These results indicate that under the current experimental setup, Transformer variants exhibited varying abilities in capturing the complex spatiotemporal variability inherent to streamflow processes. Overall, although some Transformer-based models displayed advantages over LSTM in specific metrics (such as FHV and KGE), their varying degrees of robustness suggest remaining challenges posed by spatial heterogeneity and temporal non-stationarity in hydrology modeling.

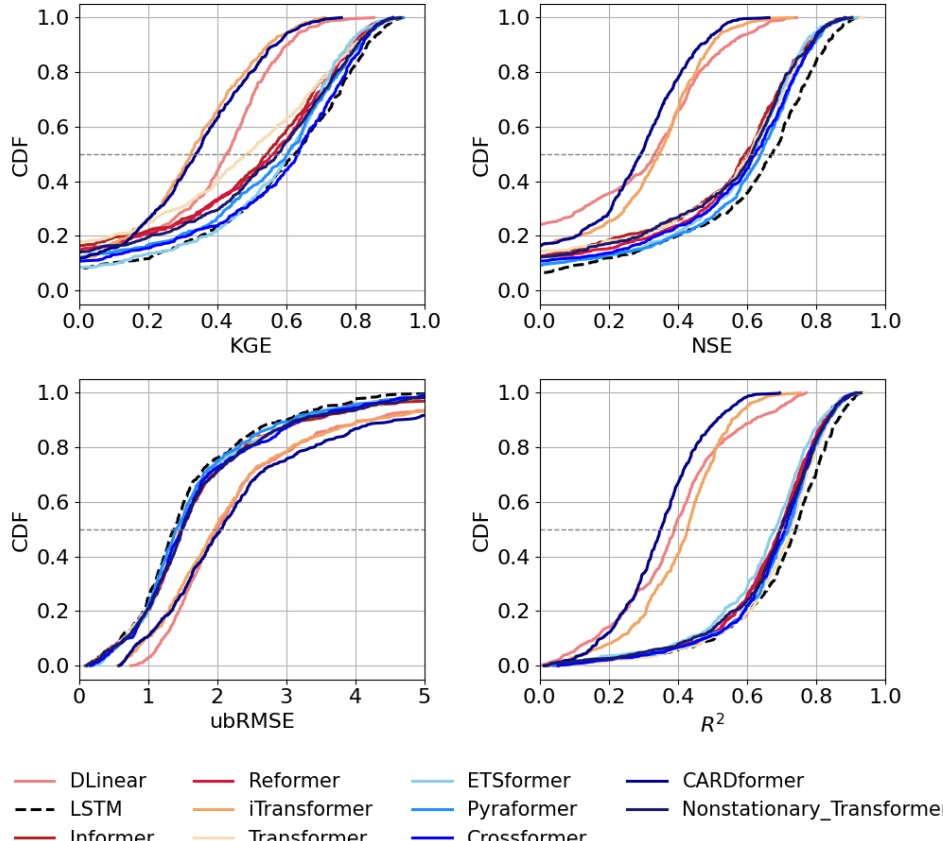

**Figure 4. Comparative analysis of model Cumulative Density Functions (CDFs) based on spatial cross-validation experiment results. The CAMELS dataset was divided into three folds according to the spatial distribution of the basins. One fold was used as the test set, while the remaining two served as the training set. By cycling through this process, every basin was evaluated as part of the test set. Combined results from all three folds were then used to compute overall evaluation metrics. Horizontal dashed lines indicate median values (CDF = 0.5). Detailed numerical results are provided in Supplementary Table S5.**

### 3.5. Zero-shot predictions

The previous experiments were based on supervised learning, involving training separate models for each variable or dataset. In contrast, recent advances in natural language processing have produced foundation models capable of zero-shot learning (Wang et al., 2019). These models can perform forecasting without relying on large, labeled datasets, by leveraging generalizable knowledge acquired during their pre-training process. To evaluate the feasibility of zero-shot predictions for hydrologic forecasting, we randomly selected seven basins from the CAMELS dataset, partly to limit the API costs. We conducted forecasting experiments for one year, providing the models with only 90 days of historical runoff data as inputs, and predicting runoff for horizons of 7, 30, and 60 days (Fig. 5, Supplementary Fig. S3, Table S6). As a benchmark for these zero-shot experiments, we also trained an LSTM model using supervised learning on the same set of basins. At the shortest forecasting horizon of 7 days, LSTM achieved a KGE of 0.50. In comparison, GPT-3.5, Llama 3 8B, and TimeGPT achieved KGE values of 0.53, 0.54, and 0.68 respectively, each surpassing LSTM's performance. However, as the forecast horizon increased, all model performances dropped, with LSTM's KGE dropping below 0.15 at 30 and 60 days. Notably, TimeGPT maintained relatively robust performance at the 30-day horizon, achieving a KGE value of 0.33.

We acknowledge that pre-trained LLMs may not genuinely capture the physical relationships among hydrologic variables without fine-tuning, as their forecasting ability relies predominantly on learned contextual patterns rather than hydrological causality. Additional exploratory experiments using the ChatGPT API on regression and forecasting tasks resulted in very low or negative NSE and KGE values, confirming that pre-trained LLMs struggle to infer hydrologic relationships without domain-specific fine-tuning.

Nevertheless, these results are surprisingly promising, suggesting that large-scale pretrained models originally developed for text or other generic time series tasks might provide credible hydrologic forecasts without domain-specific data. This study represents an initial attempt to employ LLMs and TSAMs for zero-shot hydrologic forecasting. However, even the best-performing model (TimeGPT) still struggled to accurately capture peak flow events (Fig. 5). Future work should focus on improving these models' forecasting capabilities through strategies such as prompt engineering, domain adaptation, or fine-tuning. Overall, by introducing advanced self-supervised methods, our findings highlight a promising direction for hydrologic predictions in data-limited regions.

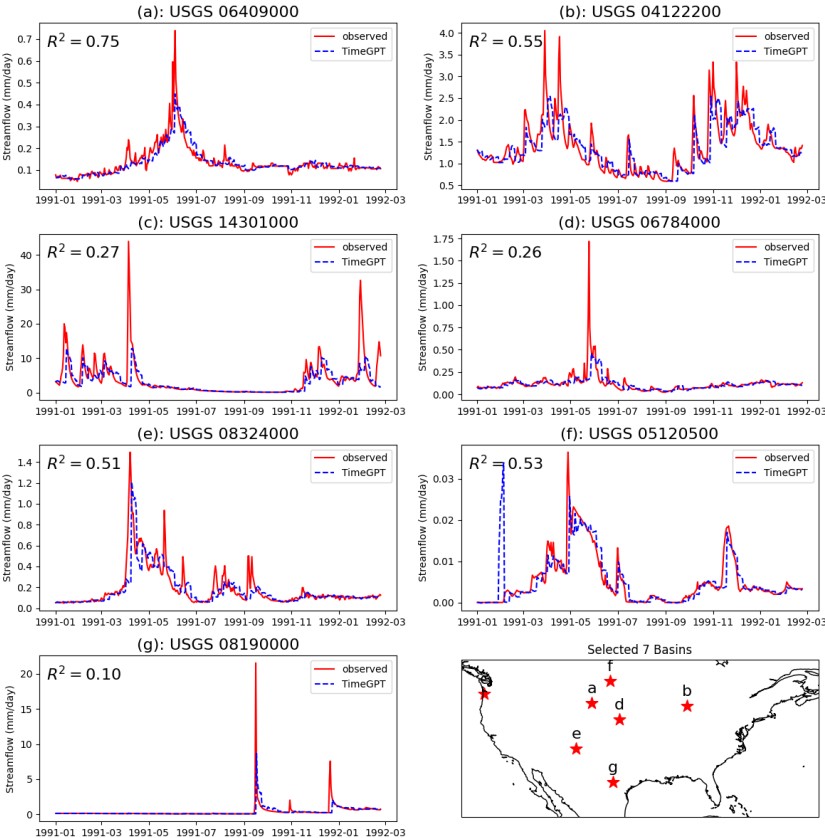

**Figure 5. Observed and predicted runoff time series for seven randomly-selected basins. The red line represents observed data, while the blue dashed line indicates predictions from TimeGPT. A 90-day historical data window was used to forecast the subsequent 7-day period. Detailed numerical results are provided in Supplementary Table S6.**

**3.6 Further discussion**

As shown in Section 3.1, Transformers often underperformed compared to LSTM in regression tasks, possibly due to several inherent characteristics. Firstly, the self-attention mechanism in Transformers is permutation invariant, meaning it does not inherently capture sequential dependencies. Although positional encoding brings some temporal context, it remains insufficient for modeling medium-term to long-term trends or periodic fluctuations (Zeng et al., 2022). Secondly, hydrologic time series data are effectively low-semantic, continuous signals with substantial noise, conditions under which Transformer models are susceptible to overfitting (Zeng et al., 2022). Similar limitations have been observed in other research fields. For example, Transformers in reinforcement learning (RL) demonstrated sensitivity to initialization parameters, complicating the learning of Markov processes (Parisotto et al., 2020). In computer vision, Convolutional Neural Networks (CNNs) can capture global contextual features  similar to Transformers by enlarging convolutional kernels or utilizing patch-based processing (Wang et al., 2023b). In contrast, LSTM leverages gating mechanisms to selectively store or discard information over sequential steps, thus typically achieving better prediction in regression tasks.

Transformers show an advantage over LSTM in autoregressive tasks, however, especially when predicting longer horizons. Previous studies have similarly reported that Transformers may initially underperform in short-term forecasting due to the global nature of attention mechanisms, which makes them less sensitive to immediate local fluctuations. However, their performance improves over medium- to long-term horizons (more than 7 days) (Pölz et al., 2024; Wang et al., 2023a). For example, Wang et al. (2024) evaluated soil moisture forecasting over time lags of 1, 3, 5, 7, and 10 days, and observed that the performance of LSTM decreased as the forecasting horizon increased. They attributed Transformers' robustness in long-term prediction tasks to their greater capacity for modeling complex, nonlinear patterns in noisy environments.

As model scale and datasets grow, computational demands and associated environmental impacts also increase. To assess these impacts, we quantified energy consumption and carbon dioxide emissions using an NVIDIA A100 SXM4 80GB GPU. By measuring the training time over a complete training cycle (30 epochs), we estimated carbon footprints using a local electricity carbon emission factor of 0.432 kg/kWh (Supplementary Table S7) (Lacoste et al., 2019). For example, training the LSTM model on the CAMELS dataset required about one hour and resulted in approximately 0.17 kg of $CO_2$ equivalent emissions. In comparison, the Non-stationary Transformer produced approximately 0.62 kg of $CO_2$ eq., roughly three times higher than LSTM, while Crossformer emitted approximately 2.25 kg of $CO_2$ eq., about thirteen times higher than LSTM. Although attention-based models exhibit higher energy consumption for individual tasks, large pre-trained foundation models could potentially offer significant efficiency gains if applied to multiple downstream tasks without retraining from scratch. For example, the pre-trained foundation models can support a lot of applications simultaneously and even surpass LSTM in zero-shot forecasting scenarios.

Despite the promising results presented earlier, several challenges remain unresolved. Firstly, inherent limitations of models like LSTM and Transformers become evident as hydrologic modeling tasks grow more complex, making it difficult for a single model to consistently perform optimally across all scenarios. Secondly, there remains a critical gap in theoretical analysis, and rigorous mathematical studies are needed to fully understand the

fundamental reasons behind the performance differences observed between Transformer and LSTM models. Such explorations are beyond the scope of this study and will be addressed in future research. Thirdly, increasing model scales leads to higher computational costs and environmental impacts, raising important concerns regarding the balance between model capabilities and computational efficiency. Finally, it remains an open question whether existing models can effectively support more complex hydrologic tasks, such as data assimilation and integration with physical models, requiring further investigation.

## 4. Conclusions

This study introduced and evaluated a multi-task deep learning framework designed to benchmark and compare various model architectures. Through experiments conducted using multi-source and multi-scale datasets, we revealed performance differences influenced by task complexity, forecasting horizon, data availability, and regional characteristics. The LSTM model generally outperformed attention-based models in regression tasks and short-term forecasting, benefiting from its gating mechanism which manages sequential dependencies and reduces overall prediction errors. Attention-based models demonstrated advantages in capturing extreme hydrologic events and excelled in long-term autoregressive forecasting tasks. Additionally, this research explored the application of pre-trained Large Language Models (LLMs) and Time Series Attention Models (TSAMs) in zero-shot hydrologic forecasting scenarios. Without domain-specific fine-tuning, these models exhibited competitive predictive capabilities, surpassing supervised LSTM benchmarks across various forecasting horizons, highlighting their strong potential in data-limited regions.

**Code and data availability**

The code for the framework can be downloaded at https://doi.org/10.5281/zenodo.15852144. The 11 attention-based models used in this study are adapted from https://github.com/thuml/Time-Series-Library. The CAMELS dataset is available from https://ral.ucar.edu/solutions/products/camels. Global streamflow records can be obtained from Beck et al. (2020). Global soil moisture and forcing data can be downloaded from https://ismn.earth/en/dataviewer/ and from Liu et al. (2023). Snow water equivalent data and processing can be found in Song et al. (2024b). Dissolved oxygen data can be accessed via Zhi et al. (2021).

**Author Contributions**

JL conceived the study and conducted most of the experiments. During the writing process, CS offered suggestions on the scientific questions addressed by the study. FD provided suggestions regarding the pre-trained time series model. YS contributed the snow water equivalent data and offered suggestions on the early manuscript structure. WZ provided the DO data and gave feedback on the early manuscript structure. HB supplied the global streamflow dataset. TB assisted with coding for one of the zero-shot experiments. The manuscript was edited by JL, CS, FD, YS, KL, TB, HB, and NK.

**Competing interests**

Kathryn Lawson and Chaopeng Shen have financial interests in HydroSapient, Inc., a company which could potentially benefit from the results of this research. These interests have been reviewed by The Pennsylvania State University in accordance with its individual conflict of interest policy for the purpose of maintaining the objectivity and the integrity of research.

**Acknowledgements**

We thank the Global Runoff Data Centre (GRDC) for providing the observed streamflow data and the International Soil Moisture Network (ISMN) for providing the soil moisture data. JL used AI tools to improve the grammar and fluency of the manuscript.

**Financial support**

JL was supported by the National Science Foundation Award (Award no. EAR-2221880) and the Office of Biological and Environmental Research of the U.S. Department of Energy under contract DE-SC0016605. YS and CS were partially supported by subaward A23-0271-S001 from the Cooperative Institute for Research to Operations in Hydrology (CIROH) through the National Oceanic and Atmospheric Administration (NOAA)

Cooperative Agreement (grant no. NA22NWS4320003).

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
