# Peer review of "From RNNs to Transformers: Benchmarking deep learning architectures for hydrologic prediction"

_EGUsphere, 2025_

## Referee Comment (RC1)

**Revision of: "From RNNs to Transformers: benchmarking deep learning architectures for hydrologic predictions" by Jiangtao Liu et al.**

June 12, 2025

**1 General comments**

The manuscript "From RNNs to Transformers: benchmarking deep learning architectures for hydrologic predictions" by Jiangtao Liu et al. compares various deep learning models (from LSTM over transformers to LLMs) for estimating and forecasting various hydrological variables like runoff, soil moisture, snow water equivalent, and dissolved oxygen. The models are tested for five different tasks: regression, data integration, autoregression, spatial cross-validation, and zero-shot. The results show that LSTM often performs best on regression tasks, while attention-based models are getting better for more complex tasks.

This manuscript provides very useful insights for hydrological modelling using deep learning algorithms, presents the results in a comprehensive way, and is well-written. However, the methods are not yet sufficiently described to make it clear how useful the results are. Thus, I recommend reconsidering the manuscript for publication after the methods section has been adjusted. The points that need to be addressed are described below.

**2 Specific comments**

As mentioned above, the only major point to address is a more elaborate description of the methods used in this manuscript. Without a clear description of the methods, it is hard to evaluate the usefulness of the results and reproducibility is not given. Specifically, I would need more information about:

- test-train splitting

- did you do hyperparameter-tuning or how did you decide on the hyperparameters, resp. on the number of hidden layers, nodes etc.

- how are the point datasets used? Are they interpolated to raster format or are they used as predictors as points? Is the lat/lon information of the points used in the model too?

- how is the data extracted? Average over whole catchment or all pixels per catchment added to model?

- what is the temporal resolution of the input data (resampled to daily or weekly or used as is?)

- how are the LLMs used for hydrologic modelling? Just providing the data and asking them for the target variable?

Some more minor points to address are the following:

- line 16-21: the different model setups do not get clear from the abstract. This part needs to be reformulated to clarify which parameters are estimated, which tasks are done (regression, zero-shot etc.) and which models are used for these tasks.

- line 122: how did you decide on the thresholds to exclude basins larger than 5000 and smaller than 50km2?

- line 123: what type of manual quality checks did you do?

- line 135: a table with all predictors and datasets would help.

- line 148: are all other DL tools more efficient? A comparison of the calculation time of all models would be interesting (maybe in the supplementary files).

- line 165: for me, the term forecasting would be more intuitive. Or why did you choose the term data integration?

- line 180ff: Is this the same setup as Kratzert et al 2021 used? If so, it would be interesting to show how your model results perform in comparison to theirs.

- line 190ff: would it be possible to finetune the LLMs to the task of hydrologic modelling?

- line 220: Is it fine with the journal to have a combined results and discussion section?

- line 226: justify the statement that LSTM reach their performance ceiling

- Fig. 2: has wrong y axis lables. The scores should be on the x axis. Also, what is it the CDF from? Please also add the mean for an easier comparison.

- line 302: why do you use for every validation type another variable? It would be easier to focus on one variable.

- Fig. 3: Same as for Fig. 2. These results are hardly comparable like this.

**3   Technical corrections**

- line 22ff: please also mention in brackets the performance metrics (e.g. NSE) after mentioning which model performs best and write how much better this is than the other models.

- line 110f: you write 5 datasets but mention only 4.

- line 126: please also cite the ISMN paper from Dorigo et al..

- Table 1: include also LSTM

- line 165ff: Also mention the lead times for which you do the forecasting here.

- line 218: add citations

- line 247: ISMN has not been mentioned before, please do so with the citation mentioned above.

- line 305: Please add Fig S3 and Tab S3 in the paper and not in the supplementary files.

- line 314: please add R2 or other metric to text here when mentioning that Pyraformer performs best.

- Fig. 1: Please take LSTM as first bar in the plots, since it is the baseline.

- Fig. 4: Please add NSE or other metric to plot

**4   Review Criterias**

*Scientific significance: Does the manuscript represent a substantial contribution to scientific progress within the scope of Hydrology and Earth System Sciences (substantial new concepts, ideas, methods, or data)?*

(1) Yes the manuscript represents a substantial contribution

*Scientific quality: Are the scientific approach and applied methods valid? Are the results discussed in an appropriate and balanced way (consideration of related work, including appropriate references)?*

This point cannot be validated yet, more information about the methods are required, as pointed out in my comments.

*Presentation quality: Are the scientific results and conclusions presented in a clear, concise, and well-structured way (number and quality of figures/tables, appropriate use of English language)?* (2) The text is well written and the conclusions are clear and concise. The figures can still be improved, as pointed out in my comments.

**4.1 Further review points**

1. *Does the paper address relevant scientific questions within the scope of HESS?* Yes

2. *Does the paper present novel concepts, ideas, tools, or data?* Yes

3. *Are substantial conclusions reached?* Yes

4. *Are the scientific methods and assumptions valid and clearly outlined?* No, this is not clear yet, as pointed out above.

5. *Are the results sufficient to support the interpretations and conclusions?* Yes, although, this is also dependent on the last point, so further info on methods required to validate this.

6. *Is the description of experiments and calculations sufficiently complete and precise to allow their reproduction by fellow scientists (traceability of results)?* No, not yet.

7. *Do the authors give proper credit to related work and clearly indicate their own new/original contribution?* Yes

8. *Does the title clearly reflect the contents of the paper?* Yes

9. *Does the abstract provide a concise and complete summary?* Some improvements could be done to make it easier understandable. I left some comments for that in the the minor comments

10. *Is the overall presentation well structured and clear?* Yes

11. *Is the language fluent and precise?* Yes

12. *Are mathematical formulae, symbols, abbreviations, and units correctly defined and used?* Yes

13. *Should any parts of the paper (text, formulae, figures, tables) be clarified, reduced, combined, or eliminated?* Yes, as indicated above.

14. *Are the number and quality of references appropriate?* Yes

15. *Is the amount and quality of supplementary material appropriate?* Yes

---

## Author Comment (AC1)

*Reviewer #1*

**General comments**

The manuscript "From RNNs to Transformers: benchmarking deep learning architectures for hydrologic predictions" by Jiangtao Liu et al. compares various deep learning models (from LSTM over transformers to LLMs) for estimating and forecasting various hydrological variables like runoff, soil moisture, snow water equivalent, and dissolved oxygen. The models are tested for five different tasks: regression, data integration, autoregression, spatial cross-validation, and zero-shot. The results show that LSTM often performs best on regression tasks, while attention-based models are getting better for more complex tasks.

This manuscript provides very useful insights for hydrological modelling using deep learning algorithms, presents the results in a comprehensive way, and is well-written. However, the methods are not yet sufficiently described to make it clear how useful the results are. Thus, I recommend reconsidering the manuscript for publication after the methods section has been adjusted. The points that need to be addressed are described below.

**Specific comments**

As mentioned above, the only major point to address is a more elaborate description of the methods used in this manuscript. Without a clear description of the methods, it is hard to evaluate the usefulness of the results and reproducibility is not given. Specifically, I would need more information about:

1.  test-train splitting

    We appreciate the reviewer's suggestion. We will include a new subsection clearly describing our train-test splitting approach. This subsection will detail temporal splits to prevent information leakage and spatial cross-validation to evaluate model generalization.

2.  did you do hyperparameter-tuning or how did you decide on the hyperparameters, resp. on the number of hidden layers, nodes etc.

    We appreciate the reviewer's comment. In our revision, we will clarify that the hyperparameters for the LSTM models were selected based on previously published studies. For the Transformer-based models, we performed hyperparameter tuning using the validation subset of the CAMELS dataset, following prior recommendations.

3.  how are the point datasets used? Are they interpolated to raster format or are they used as predictors as points? Is the lat/lon information of the points used in the model too?
4.  how is the data extracted? Average over whole catchment or all pixels per catchment added to model?
5.  what is the temporal resolution of the input data (resampled to daily or weekly or used as is?)

    We appreciate the reviewer's questions. We plan to clarify these points in the revised manuscript as follows:

    - For point-scale observations (e.g., ISMN soil moisture and SWE), the observed values are directly used as model targets. Corresponding predictor variables (dynamic meteorological variables or static attributes) are extracted from gridded datasets based on their geographic coordinates, without interpolation to raster format.
    - For basin-scale datasets (CAMELS, Global Streamflow, and DO), observed values at basin outlets serve as model targets. Predictor variables are derived by averaging gridded inputs over each catchment.

- Latitude and longitude coordinates are used solely for the spatial extraction of predictor variables and are not directly input into the models.
- All input data have a daily temporal resolution, and no resampling was performed.

6. how are the LLMs used for hydrologic modelling? Just providing the data and asking them for the target variable?

We appreciate the reviewer's question. To address this, we will clarify in the manuscript how LLMs were applied to hydrologic modeling. Specifically, historical streamflow data and basin attributes will be formatted as textual prompts, and numerical predictions will be derived from the LLM responses.

Some more minor points to address are the following:
7. line 16-21: the different model setups do not get clear from the abstract. This part needs to be reformulated to clarify which parameters are estimated, which tasks are done (regression, zero-shot etc.) and which models are used for these tasks.

We appreciate the reviewer's suggestion. To clarify this point, we will revise the abstract to describe the prediction tasks evaluated (regression, forecasting, autoregression, spatial cross-validation, and zero-shot forecasting), specify the hydrologic variables estimated (e.g., runoff, soil moisture, snow water equivalent, dissolved oxygen), and identify the models employed.

8. line 122: how did you decide on the thresholds to exclude basins larger than 5000 and smaller than 50km2?

We appreciate the reviewer's question. In our revision, we will clarify that these thresholds (50 km$^2$ and 5,000 km$^2$) were adopted following Beck et al. (2020). Specifically, basins smaller than 50 km$^2$ were excluded to ensure sufficient spatial resolution for gridded meteorological inputs, while basins larger than 5,000 km$^2$ were excluded to minimize the influence of channel routing at daily timescales.

9. line 123: what type of manual quality checks did you do?

We appreciate the reviewer's question. We will clarify in the manuscript that our manual quality checks involved visually inspecting streamflow time series plots to identify problems such as prolonged flat lines, abrupt discontinuities, or insufficient data length.

10. line 135: a table with all predictors and datasets would help.

We will add a table summarizing all predictors and datasets used in this study.

11. line 148: are all other DL tools more efficient? A comparison of the calculation time of all models would be interesting (maybe in the supplementary files).

We appreciate the reviewer's question. To clearly address this point, we will include a detailed comparison of computational times for all models in the Supplementary Information. Additionally, we will discuss how the computational efficiency of PatchTST and TimesNet decreases as the number of catchments increases.

12. line 165: for me, the term forecasting would be more intuitive. Or why did you choose the term data integration?

We appreciate the reviewer's comment. We initially used the term "data integration" to maintain consistency with our previous studies. However, we agree that "forecasting" is clearer and more intuitive. Therefore, we will replace "data integration" with "forecasting" throughout the manuscript.

13. line 180: Is this the same setup as Kratzert et al 2021 used? If so, it would be interesting to show how your model results perform in comparison to theirs.

We appreciate the reviewer's question. In the revised manuscript, we will clarify that our benchmark model is an LSTM with a setup closely matching that of Kratzert et al. (2021). Prior to conducting our main experiments, we confirmed that our benchmark LSTM achieves similar performance (median KGE $\approx$ 0.80), thereby indirectly enabling comparison with the results reported by Kratzert et al.

14. line 190: would it be possible to finetune the LLMs to the task of hydrologic modelling?

We appreciate the reviewer's suggestion. Although fine-tuning LLMs specifically for hydrologic modeling was beyond the original scope of this study, we recognize its potential value. In the revised manuscript, we will highlight fine-tuning approaches, such as prompt-based fine-tuning and parameter-efficient methods, as promising directions for future research.

15. line 220: Is it fine with the journal to have a combined results and discussion section?

We appreciate the reviewer's suggestion. We will verify the journal's guidelines and check recent HESS publications to confirm whether a combined "Results and Discussion" section is acceptable.

16. line 226: justify the statement that LSTM reach their performance ceiling

We appreciate the reviewer's comment. We will provide a more detailed explanation by referencing recent studies, which showed minimal performance differences between LSTM and Transformer models, suggesting inherent data limitations rather than model constraints.

17. Fig. 2: has wrong y axis lables. The scores should be on the x axis. Also, what is it the CDF from? Please also add the mean for an easier comparison.

We appreciate the reviewer's suggestion. In our revision, we will correct Fig. 2 by placing the evaluation metrics (scores) on the x-axis and the cumulative density functions (CDF) on the y-axis. Additionally, we will clarify the meaning of the CDF in the figure caption and include mean values for easier comparison. Detailed numerical values will also be provided in the Supplementary Information.

18. line 302: why do you use for every validation type another variable? It would be easier to focus on one variable.

We appreciate the reviewer's question. In the revised manuscript, we will state that only the regression task utilized multiple hydrological variables (runoff, soil moisture, snow water equivalent, and dissolved oxygen) to assess model generalization. In contrast, all other tasks were conducted using the CAMELS dataset for streamflow prediction. We will highlight this distinction to clarify our methodological choices.

19. Fig. 3: Same as for Fig. 2. These results are hardly comparable like this.

We appreciate the reviewer's suggestion. To address this concern, we will improve the comparability of Fig. 3 by adding horizontal dashed reference lines at CDF = 0.5 and including numerical median values in the Supplementary Material.

3 Technical corrections

20. line 22ff: please also mention in brackets the performance metrics (e.g. NSE) after mentioning which model performs best and write how much better this is than the other models.

We appreciate the reviewer's suggestion. We will revise the abstract to include performance metrics (e.g., KGE) in parentheses when indicating which model performs best, and we will specify how much better it is compared to the other models.

21. line 110f: you write 5 datasets but mention only 4.

We appreciate the reviewer's suggestion. We will revise the manuscript to clearly list all five datasets: two runoff datasets (CAMELS and Global Streamflow), as well as datasets for soil moisture (ISMN), snow water equivalent (SWE), and dissolved oxygen (DO).

22. line 126: please also cite the ISMN paper from Dorigo et al..

We appreciate the reviewer's suggestion and will include the ISMN reference (Dorigo et al.) in the manuscript.

23. Table 1: include also LSTM

We appreciate the reviewer's suggestion and will revise Table 1 to include the LSTM model.

24. line 165ff: Also mention the lead times for which you do the forecasting here.

We appreciate the reviewer's suggestion. In our revision, we will specify that forecasts are evaluated at lead times of 1, 7, 30, and 60 days.

25. line 218: add citations

We appreciate the reviewer's suggestion. We will add appropriate citations in the revised manuscript.

26. line 247: ISMN has not been mentioned before, please do so with the citation mentioned above.

We appreciate the reviewer's suggestion. We will revise the manuscript to introduce the International Soil Moisture Network (ISMN), including the appropriate citation, before its first mention.

27. line 305: Please add Fig S3 and Tab S3 in the paper and not in the supplementary files.

We appreciate the reviewer's suggestion. We will move Fig. S3 to the main manuscript to improve readability. However, due to space limitations and the large size of Table S3, we prefer to keep it in the supplementary materials.

28. line 314: please add R2 or other metric to text here when mentioning that Pyraformer performs best.

We appreciate the reviewer's suggestion. We will revise the manuscript to include performance metrics (e.g., KGE) in the text when mentioning Pyraformer's results.

29. Fig. 1: Please take LSTM as first bar in the plots, since it is the baseline.

We appreciate the reviewer's suggestion and will revise Fig. 1 to place LSTM as the first bar in each plot.

30. Fig. 4: Please add NSE or other metric to plot

We appreciate the reviewer's suggestion and will add the $R^2$ metric to Fig. 4.

---

## Author Comment (AC2)

*Reviewer #2*

The authors evaluate different transformer models under different scenarios (i.e., regression, data integration, and autoregression) in comparison with benchmark models, LSTM networks and DLinear, using various hydrologic datasets. In addition, they compare the performance of LSTM networks with pre-trained LLM in the zero-shot forecasting for autoregression tasks. They show that LSTM networks outperform transformers in regression and data integration tasks, and attention-based methods surpass LSTM networks in autoregression and zero-shot forecasting. The paper is well written, with deep discussion. I have minor comments below.

For boarder audiences with minor ML/DL backgrounds, it would be helpful to provide brief introductions of the DL models used in the manuscript. The information can be provided in the SI if there are limited spaces in the main text. Table 1 provides the main features of different variants of transformers. Since ML/DL has many unique terms, simply providing the names of features does not really help understand their differences. Maybe the authors can adapt the table with more general features, such as "Trained on time series".

We appreciate the reviewer's suggestion. To improve accessibility for broader audiences with limited ML/DL backgrounds, we will:

- Include brief and intuitive descriptions of each deep learning model in the Supplementary Information. These descriptions will explain the key principles, and features, helping readers clearly understand their differences.
- Revise Table 1 in the main manuscript to highlight more general and easily understandable characteristics, making distinctions between the models clearer for readers without extensive ML/DL expertise.

In Section 2.3-attention models, it is better to mention that the authors use pre-trained LLM for zero-shot forecasting. This information come out in Section 2.4.5. In addition, for zero-shot forecasting, the authors only compare DL model performance in autoregression tasks, and state that LSTM underperforms pre-trained LLMs in zero-shot forecasting. How about regression and data integration tasks? I would expect that the LLMs cannot really understand the relationship between input and output without fine-tuning.

We appreciate the reviewer's suggestion. To address this comment, we will clarify in Section 2.3 that we use pre-trained Large Language Models (LLMs) for zero-shot forecasting, referencing Section 2.4.5 for further details. Additionally, we agree with the reviewer about the limitations of pre-trained LLMs without domain-specific fine-tuning. We will mention this limitation and indicate that pre-trained LLMs primarily rely on contextual patterns rather than genuine hydrological understanding. Finally, we will conduct additional experiments on regression and data integration tasks to verify the reviewer's expectation that pre-trained LLMs struggle without fine-tuning.

Specific comments:

Some figures for equations are burr, like Equation 5.

We appreciate the reviewer's suggestion. We will retype Equation 5 and other affected equations to ensure clarity and readability.

Equation 6: stating that r is Pearson's correlation coefficient.

We appreciate the reviewer's suggestion. We will clarify Equation 6 by stating that r represents Pearson's correlation coefficient between simulated and observed values.

Equation 8: I think the equation is wrong. It should be (RMSE^2-Bias^2)^0.5.

We appreciate the reviewer's suggestion. Although our original formulation is mathematically equivalent, we agree that your proposed version is clearer and more intuitive. We will revise it accordingly.

---

## Author Response (AR1)

Dear Editor,

Thank you very much for handling our manuscript. We are grateful for the detailed and constructive feedback provided by the reviewers. Based on their valuable suggestions, we have thoroughly revised our manuscript.

The primary concerns raised by the reviewers included insufficient methodological detail, ambiguous data processing procedures, limited descriptions of deep learning models, and concerns regarding the clarity of some figures and equations. To address these issues, we have improved the Data and Models section by adding descriptions of our experimental setup, dataset splitting procedures, and model architectures. We have explained how datasets were processed and described the usage and limitations of pre-trained Large Language Models (LLMs) in zero-shot forecasting tasks. Additionally, detailed introductions of all deep learning models are now provided in the Supplementary Information. Figures have been corrected for clarity, and equations have been retyped. In accordance with the submission system's requirements, the color scheme of Fig. S5 (previously Fig. S4) has been revised to be accessible for readers with color vision deficiencies.

We believe these revisions fully address the reviewers' comments and improve the manuscript's overall quality.

Thank you again for your valuable guidance throughout this review process.

**Reviewer #1**

**General comments**

The manuscript "From RNNs to Transformers: benchmarking deep learning architectures for hydrologic predictions" by Jiangtao Liu et al. compares various deep learning models (from LSTM over transformers to LLMs) for estimating and forecasting various hydrological variables like runoff, soil moisture, snow water equivalent, and dissolved oxygen. The models are tested for five different tasks: regression, data integration, autoregression, spatial cross-validation, and zero-shot. The results show that LSTM often performs best on regression tasks, while attention-based models are getting better for more complex tasks.

This manuscript provides very useful insights for hydrological modelling using deep learning algorithms, presents the results in a comprehensive way, and is well-written. However, the methods are not yet sufficiently described to make it clear how useful the results are. Thus, I recommend reconsidering the manuscript for publication after the methods section has been adjusted. The points that need to be addressed are described below.

**Specific comments**

As mentioned above, the only major point to address is a more elaborate description of the methods used in this manuscript. Without a clear description of the methods, it is hard to evaluate the usefulness of the results and reproducibility is not given. Specifically, I would need more information about:

1. test-train splitting

   We appreciate the reviewer's suggestion. To address this, we have added a new subsection (2.6 Experimental setup and hyperparameter configuration) detailing the dataset splitting methods and hyperparameter tuning procedures to enhance reproducibility and methodological transparency.

   We have included the following information in the manuscript:

*Lines 250-253:*
*2.6 Experimental setup and hyperparameter configuration*
*In the temporal experiments, each dataset was temporally divided into training, validation, and testing subsets (Table 1) to avoid information leakage. Additionally, to assess the model's spatial generalization capability, we conducted spatial cross-validation experiments.*

2. did you do hyperparameter-tuning or how did you decide on the hyperparameters, resp. on the number of hidden layers, nodes etc.

We appreciate the reviewer's question. Given the extensive experimental scale involving multiple models and datasets, we adopted previously used hyperparameters for the LSTM model from prior hydrological modeling studies (Feng et al., 2020, 2024; Kratzert et al., 2021; Liu et al., 2022a, 2023, 2024a; Song et al., 2024; Zhi et al., 2021). Specifically, we followed the configurations recommended in these studies, including the number of hidden layers, hidden units, learning rates, dropout rates, and batch sizes.

For Transformer-based models, we conducted hyperparameter tuning using the validation subset of the CAMELS dataset, guided by recommendations from the Time Series Library (Wang et al., 2024; Wu et al., 2022). Key hyperparameters tuned included the number of encoder layers, attention heads, embedding dimensions, learning rates, and dropout rates.

To address the reviewer's comment, the manuscript has been revised to include the following details:

*Lines 258-263:*
*Given the extensive experimental scale involving multiple models and datasets, we adopted previously used hyperparameters for the LSTM model from prior hydrological modeling studies (Feng et al., 2020, 2024; Kratzert et al., 2021; Liu et al., 2022a, 2023, 2024a; Song et al., 2024; Zhi et al., 2021). For Transformer-based models, hyperparameter tuning (e.g., number of encoder layers, attention heads) was conducted using the validation subset of the CAMELS dataset, guided by recommendations from the Time Series Library (Wang et al., 2024; Wu et al., 2022).*

3. how are the point datasets used? Are they interpolated to raster format or are they used as predictors as points? Is the lat/lon information of the points used in the model too?
4. how is the data extracted? Average over whole catchment or all pixels per catchment added to model?
5. what is the temporal resolution of the input data (resampled to daily or weekly or used as is?)

We appreciate the reviewer's suggestions. We have revised the manuscript accordingly to describe how point-based and basin-scale datasets were processed. Specifically, we explained that for point-based observations (ISMN soil moisture and SWE), the dynamic and static predictors were directly extracted from the gridded datasets based on their geographic coordinates. In contrast, for basin-scale datasets (CAMELS, Global Streamflow, and DO datasets), predictors were obtained by spatially averaging gridded data across the respective catchment areas. Additionally, we clarified that latitude and longitude coordinates were used only for the spatial extraction of predictors and not directly as input features in model training, except for the SWE dataset, where latitudes were included as input features.

To address the reviewer's comment, the manuscript has been revised to include the following details:

*Lines 147-155:*
*All datasets used in this study have a daily temporal resolution. For the CAMELS and Global Streamflow datasets, the target variables are daily observed streamflow at basin outlets. Dynamic*

*forcings (e.g., precipitation, temperature) and static attributes (e.g., elevation, soil properties) were obtained by spatially averaging gridded data over each catchment. For the ISMN (soil moisture) and SWE datasets, target variables are point-based observations, and their corresponding dynamic and static predictors were directly extracted from gridded datasets at the geographic coordinates of each observation site. The DO dataset, similar to CAMELS, contains target variables measured at basin outlets, with basin-averaged dynamic and static inputs derived from gridded data. Latitude and longitude coordinates were used solely for spatial extraction of predictors and were not included as direct input features for model training, except for latitude in the SWE dataset.*

6. how are the LLMs used for hydrologic modelling? Just providing the data and asking them for the target variable?

We appreciate the reviewer's suggestion. To clarify the application of LLMs in hydrologic modeling, we have described the procedure in the revised manuscript. Specifically, historical hydrological data and basin attributes were formatted into structured textual prompts, which were then provided to the LLMs to directly query predictions. The detailed prompt structure and numerical extraction methods are presented in the Supplementary Material (Text S2).

We have added the following to the manuscript:

*Lines 219-223:*
*Specifically, historical streamflow observations (e.g., previous 90 days) and basin characteristics, such as geographic coordinates and catchment area, were converted into structured textual prompts provided to the LLMs. The textual outputs from these models were then automatically parsed into numerical forecasts using regular expression extraction (Supplementary Text S2).*

Some more minor points to address are the following:
7. line 16-21: the different model setups do not get clear from the abstract. This part needs to be reformulated to clarify which parameters are estimated, which tasks are done (regression, zero-shot etc.) and which models are used for these tasks.

We appreciate the reviewer's suggestion. We have modified our abstract as follows:

*Lines 16-23:*
*The proposed framework evaluates deep learning models across diverse hydrologic prediction tasks, including regression (daily runoff, soil moisture, snow water equivalent, and dissolved oxygen prediction), forecasting (using lagged hydrologic observations combined with meteorological inputs), autoregression (forecasting based solely on historical observations), spatial cross-validation (assessing model generalization to ungauged regions), and zero-shot forecasting (prediction without task-specific training data). Specifically, we benchmarked 11 Transformer-based architectures against a baseline Long Short-Term Memory (LSTM) model and further evaluated pretrained Large Language Models (LLMs) and Time Series Attention Models (TSAMs) regarding their capabilities for zero-shot hydrologic forecasting.*

8. line 122: how did you decide on the thresholds to exclude basins larger than 5000 and smaller than 50km2?

We appreciate the reviewer's suggestion. The thresholds (50 $km^2$ and 5,000 $km^2$) were adopted directly from Beck et al. (2020). According to Beck et al. (2020), basins smaller than 50 $km^2$ were excluded to ensure that gridded meteorological forcing datasets had sufficient spatial resolution to accurately represent catchment-scale processes. Basins larger than 5,000 $km^2$ were excluded to

minimize the impacts of significant channel routing and reservoir operations that are difficult to accurately capture with daily-scale data. We have clarified this in the manuscript.

The revised manuscript reads as follows:

*Lines 124-127:*
*After excluding basins with incomplete daily records and non-reference basins, Beck et al. (2020) retained catchments ranging between 50 km² (to ensure sufficient spatial resolution of gridded meteorological inputs) and 5,000 km² (to minimize channel routing and reservoir operation effects at daily scales), resulting in a total of 4,299 basins.*

9. line 123: what type of manual quality checks did you do?

We appreciate the reviewer's suggestion. We conducted manual quality checks by visually inspecting the streamflow time series plots for each basin. Specifically, basins exhibiting clear data issues, such as flat lines (constant values), abrupt discontinuities indicating measurement errors, or insufficient record lengths (e.g., only a few months), were excluded from the analysis.

The revised manuscript is as follows:

*Lines 127-130:*
*Based on that subset, we conducted additional manual quality checks by visually inspecting streamflow time series for each basin. Basins exhibiting flat lines, abrupt discontinuities, or very short records (spanning only a few months) were excluded, ultimately retaining 3,434 basins for analysis.*

10. line 135: a table with all predictors and datasets would help.

We appreciate the reviewer's suggestion. We have added "Table 2. Overview of datasets" to the manuscript.

| Dataset | Period (Train / Validation / Test) | Target Variable | Dynamic Variables | Static Variables |
|---|---|---|---|---|
| CAMELS | 1999-10-01 to 2008-09-30 / 1980-10-01 to 1989-09-30 / 1989-10-01 to 1999-09-30 | Streamflow | precipitation, solar radiation, maximum temperature, minimum temperature, vapor pressure (NLDAS, Maurer, Daymet) | elevation, slope, area, forest fraction, LAI, GVF, soil depth, porosity, conductivity, water content, soil texture fractions (sand, silt, clay), geology, climate indices (mean precipitation, PET, aridity, snow fraction, precipitation frequency and duration extremes) |

| | | | | |
|---|---|---|---|---|
| Global Streamflow | 1999-01-01 to 2016-12-31 / 1998-01-01 to 1998-12-31 / 1980-01-01 to 1997-12-31 | Streamflow | precipitation, potential evapotranspiration, maximum temperature, minimum temperature | mean precipitation, seasonality precipitation, snow fraction, snowfall fraction, mean temperature, NDVI, elevation, slope, aspect, soil texture fractions (sand, silt, clay), soil depth, geology, forest fraction, grassland fraction, soil erosion, area |
| ISMN | 2017-01-01 to 2020-12-31 / - / 2015-04-01 to 2016-12-31 | Soil Moisture | soil temperature, surface pressure, solar radiation, air temperature, evaporation, wind speed components, volumetric soil water, precipitation | elevation, slope, aspect, soil texture (sand, clay, silt, bulk density), land surface temperature, albedo, landcover, NDVI, profile curvature, roughness, soil moisture |
| SWE | 2001-01-01 to 2015-12-31 / - / 2016-01-01 to 2019-12-31 | Snow Water Equivalent | precipitation, maximum temperature, minimum temperature, solar radiation, wind speed, humidity | latitude, elevation, slope, aspect, land cover, forest fraction, root depth, soil depth, porosity, permeability |
| DO | 1980-01-01 to 2000-12-31 / - / 2001-01-01 to 2014-12-30 | Dissolved Oxygen | precipitation, solar radiation, maximum temperature, minimum temperature, vapor pressure, streamflow | elevation, slope, area, forest fraction, LAI, GVF, soil depth, porosity, conductivity, water content, soil texture fractions (sand, silt, clay), geology, climate indices (mean precipitation, PET, aridity, snow fraction, precipitation frequency and duration extremes) |

11. line 148: are all other DL tools more efficient? A comparison of the calculation time of all models would be interesting (maybe in the supplementary files).

We appreciate the reviewer's suggestion. A comparison of computational times for all models has already been included in Table S7 of the Supplementary Information.

The revised text in the manuscript is:

*Lines 169-172:*
*It is important to note that although the computational speed of PatchTST and TimesNet is acceptable when applied to a small number of catchments or stations, our experiments revealed that their training time increases dramatically as the number of basins increases, especially for regression tasks. A detailed comparison of computational times across all models is provided in the Supplementary Information (Table S7).*

12. line 165: for me, the term forecasting would be more intuitive. Or why did you choose the term data integration?

We appreciate the reviewer's suggestion. Initially, we adopted the term "data integration" for consistency with our previous studies (Fang et al., 2018; Fang & Shen, 2020), in which it specifically described the approach of combining historical hydrological observations with future meteorological inputs to enhance forecasting accuracy. However, we agree with the reviewer that the term "forecasting" is more intuitive. Thus, we have uniformly replaced the term "data integration" with "forecasting" throughout the manuscript.

13. line 180: Is this the same setup as Kratzert et al 2021 used? If so, it would be interesting to show how your model results perform in comparison to theirs.

We appreciate the reviewer's suggestion. We assume that the reviewer refers to the study by Kratzert et al. (2021) published in Hydrology and Earth System Sciences, which primarily employed an LSTM model similar to the baseline used in our study. Before conducting the main experiments, we verified that our baseline LSTM achieved a performance level (median KGE $\approx 0.80$) comparable to that reported by Kratzert et al. (2021). Since the Transformer-based models in our study were benchmarked directly against this validated LSTM baseline, our results indirectly provide a meaningful comparison with those of Kratzert et al. (2021).

We have added the following to the manuscript:

*Lines 253-256:*
*Our baseline LSTM configuration closely follows that of Kratzert et al. (2021). Prior to our main experiments, we confirmed that this baseline LSTM achieved a performance (median KGE $\approx 0.80$) comparable to that reported by Kratzert et al. (2021), thus providing indirect validation and enabling comparability with established benchmarks.*

14. line 190: would it be possible to finetune the LLMs to the task of hydrologic modelling?

We appreciate the reviewer's suggestion. Our study aimed primarily to assess the zero-shot forecasting capabilities of pretrained Large Language Models (LLMs) without additional task-specific fine-tuning. Fine-tuning LLMs specifically for hydrologic modeling was beyond the scope of this study. However, we agree that approaches such as prompt-based fine-tuning and parameter-efficient tuning could enhance model performance. We have highlighted this promising future research direction in the revised manuscript.

We have added the following to the manuscript:

*Lines 223-225:*
*Although fine-tuning LLMs specifically for hydrologic modeling was not explored in this study, prompt-based fine-tuning, parameter-efficient tuning methods, and further exploration of repeated queries and parameter variations represent promising directions for future research.*

15. line 220: Is it fine with the journal to have a combined results and discussion section?

We appreciate the reviewer's suggestion. Based on recent articles published in this journal, a combined "Results and Discussion" section appears acceptable. We adopted this format because it allows immediate interpretation and contextualization of each set of results upon presentation.

16. line 226: justify the statement that LSTM reach their performance ceiling

We appreciate the reviewer's suggestion. The statement that LSTM models have reached their "performance ceiling" is supported by our previously published study (Liu et al., 2024), where Transformer-based architectures were systematically evaluated against LSTM models on the CAMELS dataset. This study found that despite methodological differences between Transformer-based and LSTM models, both achieved comparably high predictive performance, suggesting that further substantial improvement might be inherently constrained by existing hydrological datasets and associated data uncertainties, rather than by model architectures alone.

The revised text in the manuscript is:

*Lines 271-274:*

*This result suggests that the LSTM model may be nearing its performance ceiling, consistent with previous studies (Liu et al., 2024a; Vu et al., 2023) reporting minimal performance differences between Transformer variants and LSTM. These observations imply that additional improvements may be inherently limited by data uncertainties or intrinsic constraints of current hydrologic datasets.*

17. Fig. 2: has wrong y axis lables. The scores should be on the x axis. Also, what is it the CDF from? Please also add the mean for an easier comparison.

We appreciate the reviewer's suggestion. We have corrected Fig. 2 by placing the evaluation metrics (scores) on the x-axis and the cumulative probability (CDF) on the y-axis. Additionally, we have now clarified what the CDF represents by adding a brief explanation in the figure caption. We have also added horizontal lines at a CDF value of 0.5 to indicate the median values. Detailed numerical values corresponding to these median values are provided in the Supplementary Information (Table S2).

The revised caption for Fig. 2 is as follows:

*Figure 2. Comparative analysis of the Cumulative Density Functions (CDF) based on results from time-lagged forecasting experiments. The CDF represents the distribution of performance metrics across all basins. The four columns correspond to different evaluation metrics: Kling–Gupta Efficiency (KGE), Nash–Sutcliffe Efficiency (NSE), unbiased Root Mean Square Error (ubRMSE), and Coefficient of Determination ($R^2$). The four rows represent different time lags: 1 day, 7 days, 30*

*days, and 60 days. Horizontal lines indicate median values (CDF = 0.5). Detailed numerical results, including median values, are provided in Supplementary Table S2.*

18. line 302: why do you use for every validation type another variable? It would be easier to focus on one variable.

We appreciate the reviewer's suggestion. Indeed, our intention in the regression task was to demonstrate the model's capability to generalize across a variety of hydrological conditions and target variables (runoff, soil moisture, snow water equivalent, and dissolved oxygen). By evaluating multiple distinct hydrological variables, we aimed to illustrate the model's flexibility and robustness across diverse hydrological contexts.

However, for other tasks (such as forecasting, autoregression, and spatial validation), we focused only on streamflow prediction using the CAMELS dataset. This decision was made primarily to reduce redundancy, maintain clarity in interpreting results, and control the computational burden of extensive experiments across multiple datasets and tasks.

19. Fig. 3: Same as for Fig. 2. These results are hardly comparable like this.

We appreciate the reviewer's suggestion. To enhance the comparability and clarity of Figure 3, we have added horizontal dashed reference lines at CDF = 0.5. Additionally, the numerical median values corresponding to these reference lines are provided in Supplementary Table S5.

The revised caption for Fig. 3 is as follows:

*Figure 3. Comparative analysis of the Cumulative Density Functions (CDF) based on spatial cross-validation experiment results. The CAMELS dataset was divided into three folds according to the spatial distribution of the basins. One fold was used as the test set, while the remaining two served as the training set. By cycling through this process, every basin was evaluated as part of the test set. Combined results from all three folds were then used to compute overall evaluation metrics. Horizontal dashed lines indicate median values (CDF = 0.5). Detailed numerical results are provided in Supplementary Table S5.*

3 Technical corrections

20. line 22ff: please also mention in brackets the performance metrics (e.g. NSE) after mentioning which model performs best and write how much better this is than the other models.

We appreciate the reviewer's suggestion. We have revised the abstract to indicate performance metrics (e.g., KGE) when describing model results.

The revised text in the abstract is:

*Lines 23-25*
*Results show that LSTM models perform best in regression tasks, especially on the global streamflow dataset (median KGE = 0.75), surpassing the best-performing Transformer-based model by 0.11 KGE points.*

21. line 110f: you write 5 datasets but mention only 4.

We appreciate the reviewer's suggestion. We have revised the manuscript to list all five datasets used, clarifying that we utilized two runoff datasets (CAMELS and Global Streamflow) and three additional datasets (ISMN soil moisture, SWE, and dissolved oxygen).

The revised text in the manuscript is:

*Lines 113-115:*
*We mainly used five multi-source hydrologic datasets (Supplementary Fig. S2), including two runoff datasets—Catchment Attributes and Meteorology for Large-sample Studies (CAMELS) and Global Streamflow—as well as datasets for soil moisture (ISMN), snow water equivalent (SWE), and dissolved oxygen (DO) (Table 1).*

22. line 126: please also cite the ISMN paper from Dorigo et al..

We appreciate the reviewer's suggestion. We have added the ISMN citation (Dorigo et al.) to the revised manuscript.

23. Table 1: include also LSTM

We appreciate the reviewer's suggestion. We have included the LSTM model in Table 1 as recommended.

24. line 165ff: Also mention the lead times for which you do the forecasting here.

We appreciate the reviewer's suggestion. The revised text in the manuscript is:

*Line 192: In this study, forecasts were evaluated for lead times of 1, 7, 30, and 60 days.*

25. line 218: add citations

We appreciate the reviewer's suggestion. We have clarified that although some studies assessing other variables did not explicitly use the terms FHV/FLV, they applied conceptually similar percentile-based metrics to evaluate model performance under extreme conditions. We have added relevant citations and modified the sentence in the manuscript to:

*Lines 246-248:*

*Although initially developed for flow analysis, metrics conceptually similar to FHV/FLV have also been applied to other variables to evaluate model performance at their extreme upper and lower ranges (Bayissa et al., 2021; Brunner and Voigt, 2024).*

26. line 247: ISMN has not been mentioned before, please do so with the citation mentioned above.

We appreciate the reviewer's suggestion. The revised manuscript text is:

*Line 132: The global soil moisture dataset from the International Soil Moisture Network (ISMN)*

27. line 305: Please add Fig S3 and Tab S3 in the paper and not in the supplementary files.

We appreciate the reviewer's suggestion. We have moved Fig. S3 into the main text to enhance readability. However, due to space limitations and the size of Table S3, we have retained Table S3 in the supplementary materials.

28. line 314: please add R2 or other metric to text here when mentioning that Pyraformer performs best.

We appreciate the reviewer's suggestion. The revised manuscript text is:

*Line 362: (e.g., Pyraformer achieved an KGE of 0.15 at the 30-day horizon)*

29. Fig. 1: Please take LSTM as first bar in the plots, since it is the baseline.

We appreciate the reviewer's suggestion. We have rearranged Fig. 1 to display LSTM as the first bar in each plot.

30. Fig. 4: Please add NSE or other metric to plot

We appreciate the reviewer's suggestion. We have added the $R^2$ metric to Fig. 4 to illustrate model performance.

**Reviewer #2**

The authors evaluate different transformer models under different scenarios (i.e., regression, data integration, and autoregression) in comparison with benchmark models, LSTM networks and DLinear, using various hydrologic datasets. In addition, they compare the performance of LSTM networks with pre-trained LLM in the zero-shot forecasting for autoregression tasks. They show that LSTM networks outperform transformers in regression and data integration tasks, and attention-based methods surpass LSTM networks in autoregression and zero-shot forecasting. The paper is well written, with deep discussion. I have minor comments below.

For boarder audiences with minor ML/DL backgrounds, it would be helpful to provide brief introductions of the DL models used in the manuscript. The information can be provided in the SI if there are limited spaces in the main text. Table 1 provides the main features of different variants of transformers. Since ML/DL has many unique terms, simply providing the names of features does not really help understand their differences. Maybe the authors can adapt the table with more general features, such as "Trained on time series".

We appreciate the reviewer's suggestion. To make the manuscript more accessible to broader audiences who may not have extensive ML/DL backgrounds, we have:

a. Added detailed, reader-friendly descriptions of each deep learning model used in our study to the Supplementary Information (Text S1). These descriptions provide intuitive explanations of the main principles and features of each model, helping readers understand their differences and applications.

*Text S1. Model description*
*CARDformer is a Transformer-based deep learning model for multivariate time series forecasting. It employs channel-aligned attention to model both temporal dependencies and inter-variable correlations, overcoming the limitations of channel-independent approaches. A token blend module*

*combines information at multiple temporal resolutions to capture both local details and extended temporal dependencies.*

*Crossformer is a Transformer variant developed to capture dependencies across both time and variables in multivariate time series. It uses Dimension-Segment-Wise embedding to form a 2D time–variable representation and a Two-Stage Attention mechanism to model temporal patterns within each variable and relationships among variables, enhanced by a Hierarchical Encoder-Decoder for multi-scale forecasting.*

*ETSformer integrates exponential smoothing techniques into the Transformer framework to enhance time series forecasting. It employs Exponential Smoothing Attention (ESA) and Frequency Attention (FA) mechanisms. ESA prioritizes recent observations to capture trends, while FA captures repeating seasonal patterns using Fourier transforms. This structured attention enables ETSformer to decompose forecasts into interpretable trend and seasonal components, thereby improving both reliability and transparency of predictions.*

*Informer is designed to address the challenges in forecasting long time series. It employs a ProbSparse self-attention mechanism that selectively attends to the most important parts of the input, reducing computational complexity and memory usage compared to traditional self-attention. Informer also introduces self-attention distilling, which progressively shortens input sequences layer by layer while preserving essential information, alleviating memory limitations. Furthermore, its generative decoder produces complete forecast sequences in a single forward pass, substantially increasing inference speed and minimizing cumulative prediction errors.*

*iTransformer is a Transformer-based method that models multivariate time series by inverting the data processing dimensions. Specifically, rather than embedding all variables at each time step into a single temporal token, it treats the entire historical series of each individual variable independently as a token. This inversion enables the self-attention mechanism to capture correlations among different variables, while a shared feed-forward network separately learns representations from the full historical context of each variable.*

*The Non-stationary Transformer addresses non-stationarity, a common characteristic of real-world time series where statistical properties change over time. It employs a preprocessing step called Series Stationarization, which normalizes input data to improve predictability and ease model training. However, since this process may inadvertently eliminate valuable non-stationary dynamics, the model introduces De-stationary Attention, an attention mechanism designed to recover and reintegrate the intrinsic non-stationary temporal dependencies into the prediction.*

*PatchTST employs a patch-based approach inspired by techniques from fields such as computer vision, segmenting a long time series into smaller segments (patches). Each patch is then transformed into an input token for the Transformer model, which applies global self-attention across these patch tokens. This reduces the effective sequence length, allowing modeling of longer historical contexts.*

*Pyraformer introduces a pyramidal attention structure that models time series data at multiple resolutions, capturing local temporal dependencies through intra-scale connections at fine-grained scales and long-range dependencies via inter-scale connections at coarser scales. The method utilizes a hierarchical pyramidal attention module (PAM), combining inter-scale tree structures with intra-scale neighboring connections. This hierarchical approach shortens the path connecting distant time points, enabling modeling of long-range temporal patterns while maintaining linear computational complexity.*

*Reformer enhances traditional Transformers to handle long sequences through two key innovations. First, it introduces Locality-Sensitive Hashing (LSH) attention, which replaces standard dot-product attention by identifying the most relevant keys for each query, reducing computational complexity from quadratic (proportional to the square of sequence length) to near-linear (proportional to the sequence length multiplied by its logarithm). Second, Reformer employs reversible residual layers, which reduce memory usage by reconstructing intermediate activations during the backward pass rather than storing them at each layer.*

*TimesNet transforms one-dimensional time series data into two-dimensional representations, capturing intraperiod-variations and interperiod-variations. Leveraging the inherent multi-periodicity in real-world time series, TimesNet adaptively learns and identifies significant periodicities, reshaping data into structured 2D tensors. By doing so, TimesNet utilizes convolutional neural network modules (TimesBlock) to model complex temporal dependencies.*

*The Vanilla Transformer is the foundational model upon which many subsequent variants are built. It models sequences through attention mechanisms, specifically multi-head self-attention and encoder-decoder attention, eliminating recurrent and convolutional structures. This attention mechanism allows the Transformer to simultaneously analyze global dependencies between all sequence elements, facilitating parallel computation and faster training. Additionally, positional encodings (to capture sequence order), residual connections, and layer normalization (to stabilize training) further enhance its effectiveness in learning from sequential data.*

*The Long Short-Term Memory (LSTM) network is a type of recurrent neural network (RNN) designed to handle data sequences where long-term dependencies are crucial. LSTMs use gating mechanisms (input, forget, and output gates) that dynamically control how information is maintained, discarded, or updated at each step in the sequence.*

b.  Modified Table 2 in the main manuscript to highlight more general characteristics of each model.

| Models Name | Main Feature | General Feature | Reference |
|---|---|---|---|
| CARDformer | Channel-aligned attention; token blend module (originally referred to as CARD in the paper) | Multivariate correlation modeling | (Xue et al., 2024) |
| Crossformer | Cross-dimension dependency; dimension-segment-wise embedding; two-stage attention | Multivariate dependency modeling | (Zhang and Yan, 2022) |
| ETSformer | Exponential Smoothing Attention (ESA); Frequency Attention (FA) | Trend and seasonality modeling | (Woo et al., 2022) |
| Informer | ProbSparse self-attention; self-attention distilling; Generative style decoder | Efficient long-sequence forecasting | (Zhou et al., 2021) |

| | | | |
|---|---|---|---|
| iTransformer | Inverted Dimension; embedding the whole series as the token | Multivariate interaction modeling | (Liu et al., 2024b) |
| Non-stationary Transformer | Series Stationarization; de-stationary attention | Handling non-stationary time series | (Liu et al., 2022b) |
| PatchTST | Patch-based tokenization; channel independence | Segmentation-based time series modeling | (Nie et al., 2023) |
| Pyraformer | Pyramidal attention; hierarchical multi-resolution structure | Multi-scale temporal analysis | (Liu et al., 2021) |
| Reformer | Locality-Sensitive Hashing (LSH) attention; reversible residual layers | Efficient handling of long sequences | (Kitaev et al., 2020) |
| TimesNet | Converts 1D variations to 2D; intraperiod/interperiod analysis | Multi-periodicity pattern extraction | (Wu et al., 2022) |
| Vanilla Transformer | Multi-head self-attention; residual connections and layer normalization; positional encodings | General-purpose sequence modeling | (Vaswani et al., 2017) |
| LSTM | Gating mechanisms (input, forget, and output gates) | Captures temporal dependencies | (Hochreiter and Schmidhuber, 1997) |

In Section 2.3-attention models, it is better to mention that the authors use pre-trained LLM for zero-shot forecasting. This information come out in Section 2.4.5. In addition, for zero-shot forecasting, the authors only compare DL model performance in autoregression tasks, and state that LSTM underperforms pre-trained LLMs in zero-shot forecasting. How about regression and data integration tasks? I would expect that the LLMs cannot really understand the relationship between input and output without fine-tuning.

We appreciate the reviewer's suggestion. Following your recommendation, we have added the following content to the manuscript:

*Lines 173-175:*
*In addition, we included pre-trained Large Language Models (LLMs) for zero-shot forecasting, as described in detail in Section 2.4.5.*

We agree with the reviewer's comment that pre-trained LLMs, without domain-specific fine-tuning, might not genuinely capture the physical relationships between inputs and outputs. Their predictions are mainly based on learned contextual patterns rather than hydrological causality. We explicitly state this limitation in the manuscript:

*Lines 422-424:*
*We acknowledge that pre-trained LLMs may not genuinely capture the physical relationships among hydrologic variables without fine-tuning, as their forecasting ability relies predominantly on learned contextual patterns rather than hydrological causality.*

Additionally, in response to your question about the performance of pretrained LLMs on regression and forecasting (data integration) tasks, we conducted exploratory tests using the ChatGPT API. Due to computational costs and token limitations (each experiment costing approximately 5–10 USD), these experiments were limited in number. Our preliminary results indicated very low or negative performance (e.g., NSE and KGE values), suggesting that, without domain-specific fine-tuning, pre-trained LLMs struggle to infer explicit input-output hydrological relationships necessary for these tasks. We have added the following to the manuscript:

*Lines 424-426:*

*Additional exploratory experiments using the ChatGPT API on regression and forecasting tasks resulted in very low or negative NSE and KGE values, confirming that pre-trained LLMs struggle to infer hydrologic relationships without domain-specific fine-tuning.*

Specific comments:
Some figures for equations are burr, like Equation 5.

We appreciate the reviewer's suggestion. We have retyped all affected equations, including Equation 5.

Equation 6: stating that r is Pearson's correlation coefficient.

We appreciate the reviewer's suggestion. We have clarified Equation 6 by defining r as Pearson's correlation coefficient between simulated and observed values.

Equation 8: I think the equation is wrong. It should be (RMSE^2-Bias^2)^0.5.

We appreciate the reviewer's suggestion. We confirm that our original formula for ubRMSE is mathematically equivalent to your proposed formulation. Nonetheless, we agree that your suggested version is clearer and more intuitive for readers. Therefore, we have revised Equation 8 as suggested:

$$ubRMSE = \sqrt{RMSE^2 - Bias^2}$$

---

## Author Response (AR2)

Dear Editor,

Thank you very much for handling our manuscript.

In preparing the final version, we carefully proofread the manuscript and corrected grammatical errors and typos, improved clarity by rephrasing several sentences, and replaced some figures with clearer versions. To transparently show these minor revisions, we have uploaded an additional document highlighting all changes using the track-changes feature.

Please let us know if any further information or action is required.

Thank you again for your support throughout the review process.

Best regards,

Jiangtao Liu

[revised manuscript text omitted]

Page 9: [1] Deleted        Liu, Jiangtao        11/1/2025 11:56:00 PM